# Physical and sexual abuse in childhood and adolescence and leukocyte telomere length: A pooled analysis of the study on psychosocial stress, spirituality, and health

Erica T. Warner[1,2]*, Ying Zhang[1,3], Yue Gu[1], Tâmara P. Taporoski[4,5], Alexandre Pereira[5], Immaculata DeVivo[6], Nicholas D. Spence[1,7], Yvette Cozier[8], Julie R. Palmer[8], Alka M. Kanaya[9], Namratha R. Kandula[10], Shelley A. Cole[11], Shelley Tworoger[12], Alexandra Shields[1]

1 MGH/Harvard Center on Genomics, Vulnerable Populations, and Health Disparities, Mongan Institute, Massachusetts General Hospital and Harvard Medical School, Boston, Massachusetts, United States of America, 2 Department of Medicine, Clinical Translational Epidemiology Unit, Massachusetts General Hospital and Harvard Medical School, Boston, Massachusetts, United States of America, 3 Division of Sleep Medicine, Brigham and Women's Hospital, Harvard Medical School, Boston, Massachusetts, United States of America, 4 Department of Neurology (Sleep Medicine), Northwestern University Feinberg School of Medicine, Chicago, Illinois, United States of America, 5 Laboratory of Genetics and Molecular Cardiology, Heart Institute (Incor), University of São Paulo Medical School, São Paulo, São Paulo, Brazil, 6 Channing Division of Network Medicine, Department of Medicine, Brigham and Women's Hospital and Harvard Medical School, Boston, Massachusetts, United States of America, 7 Department of Sociology, University of Toronto, Toronto, Ontario, Canada, 8 Slone Epidemiology Center, Boston University, Boston, Massachusetts, United States of America, 9 Division of General Internal Medicine, University of California San Francisco, San Francisco, Califonia, United States of America, 10 Department of Medicine, Northwestern University, Evanston, Illinois, United States of America, 11 Department of Genetics, Texas Biomedical Research Institute, San Antonio, Texas, United States of America, 12 Department of Cancer Epidemiology, Moffitt Cancer Center and Research Institute, Tampa, Florida, United States of America

* ewarner@mgh.harvard.edu

## Abstract

### Introduction

We examined whether abuse in childhood and/or adolescence was associated with shorter telomere length in a pooled analysis of 3,232 participants from five diverse cohorts. We also assessed whether religion or spirituality (R/S) could buffer deleterious effects of abuse.

### Methods

Physical and sexual abuse in childhood (age <12) and adolescence (age 12–18) was assessed using the Revised Conflict Tactics Scale and questions from a 1995 Gallup survey. We measured relative leukocyte telomere lengths (RTL) using quantitative real time polymerase chain reaction. We used generalized estimating equations to assess associations of physical and sexual abuse with log-transformed RTL z-scores. Analyses were conducted in each cohort, overall, and stratified by extent of religiosity or spirituality and religious coping in adulthood. We pooled study-specific estimates using random-effects models and assessed between-study heterogeneity.

**Data Availability Statement:** The data from the Study on Psychosocial Stress, Spirituality, and Health (SSSH) are only available upon request due

to consent restrictions on publicly sharing data imposed by the Partners Human Research Committee (the institutional review board of Mass General Brigham) to protect participants' privacy and confidentiality. We encourage enquiry about data access; more information can be found at https://cgvh.harvard.edu/national-consortium-psychosocial-stress-spirituality-and-health or by contacting sssh@partners.org.

**Funding:** This work was supported by The John Templeton Foundation https://www.templeton.org/ [59607 to A.E.S] and the National Institutes of Health https://www.nih.gov/ [K01CA188075 to E.T. W; U01CA164974, R01CA098663, and R01CA058420 to J.R.P (Black Women's Health Study); 1R01HL093009, 2R01HL093009, R01HL120725, UL1RR024131, UL1TR001872, and P30DK098722 to A.M.K and N.R.K (MASALA); U01CA176726, R01CA163451, and R01CA67262 to Nurses' Health Study II; 75N92019D00027, 75N92019D00028, 75N92019D00029, 75N92019D00030, R01HL109315, R01HL109301, R01HL109284, R01HL109282, R01HL109319, U01HL41642, U01HL41652, U01HL41654, U01HL65520, and U01HL65521 to Strong Heart Study. The content is solely the responsibility of the authors and does not necessarily represent the official views of the National Institutes of Health or the Indian Health Service (IHS). The funders had no role in study design, data collection and analysis, decision to publish, or preparation of the manuscript.

**Competing interests:** The authors have declared that no competing interests exist.

## Results

Compared to no abuse, severe sexual abuse was associated with lower RTL z-scores, in childhood: -15.6%, 95% CI: -25.9, -4.9; p-trend = 0.04; p-heterogeneity = 0.58 and in adolescence: -16.5%, 95% CI: -28.1, -3.0; p-trend = 0.08; p-heterogeneity = 0.68. Sexual abuse experienced in both childhood and adolescence was associated with 11.3% lower RTL z-scores after adjustment for childhood and demographic covariates (95% CI: -20.5%, -2.0%; p-trend = 0.03; p-heterogeneity = 0.62). There was no evidence of effect modification by R/S. Physical abuse was not associated with telomere length.

## Conclusions

Sexual abuse in childhood or adolescence was associated with a marker of accelerated biological aging, decreased telomere length. The lack of moderation by R/S may be due to inability to capture the appropriate time period for those beliefs and practices.

## Introduction

There is increasing evidence that adverse experiences in childhood and/or adolescence, including physical, sexual, and emotional abuse and neglect [1] are associated with higher risk of multiple conditions in adulthood [2] including cardiovascular disease [3], altered stress and inflammatory responses [4–7], obesity [8], cancers [9–11] and multimorbidity [12]. Previous studies have demonstrated that exposures during these important developmental periods can have long lasting effects on physical and mental health [13]. The specific mechanisms through which these exposures affect later disease risk have not been fully explicated, but telomeres, which form protective caps at the ends of chromosomes, are a likely pathway [14].

Telomeres are repetitive DNA sequences at the ends of chromosomes that prevent physical deterioration of the chromosome during cell division [15]. Telomeres shorten with each cellular replication, are an important regulator of cellular senescence and apoptosis [16], and are considered an important biological clock, measuring aging at a cellular level [17, 18]. Factors such as older age, smoking, diet, and inflammation have been associated with shorter mean leukocyte telomere length [19–23]. Chronic stress is associated with telomere attrition [24], and telomeres are a hypothesized mechanism for how stressful exposures 'get under the skin' leading to chronic disease [25, 26].

A recent meta-analysis identified 41 studies (12 case-control, 25 cross-sectional, 4 prospective) published through July 2016 that investigated the impact of early life adversity on telomere length [27]. Early-life adversity was associated with shorter telomere length (Cohen's *d* effect size = -0.35; 95% confidence interval (CI), -0.46 to -0.24; $P$ <0.0001) with stronger associations observed in case-control and cross-sectional analyses as compared to prospective studies [28]. Importantly, this analysis did not assess moderating effects of resources for resilience, such as religion and spirituality (R/S). However, they demonstrated that the association between early-life adversity and telomere length decreased with increasing time since exposure, suggesting that telomere shortening may be reversible, or that the rate of adversity-induced shortening may be modiable [28]. We propose, and others have demonstrated [29], that R/S is an important coping resource for many individuals and may reduce the deleterious effects of early-life adversity on health [30].

In the present study, we assess the association between severity of physical and sexual abuse in childhood and/or adolescence and adult telomere length, using data from five cohorts

participating in the Study on Psychosocial Stress, Spirituality, and Health (SSSH) representing multiple racial/ethnic groups. We hypothesized that abuse in childhood and/or adolescence would be associated with shorter telomeres and that religious and spiritual (R/S) practices or beliefs in adulthood would modify the association such that telomere attrition associated with abuse would be lessened among participants with greater R/S.

## Material and methods

### Study population

SSSH, established in 2016, is designed to examine the mechanisms through which psychosocial stress contributes to disease and whether these associations are moderated or mediated by religiosity and/or spirituality. It includes a sample of participants from five US-based (Black Women's Health Study (BWHS) [31], Hispanic Community Health Study/Study of Latinos (HCHS/SOL) [32], Mediators of Atherosclerosis in South Asians Living in America (MASALA) [33], Nurses' Health Study II (NHSII) [34], Strong Heart Study (SHS) [35]) and two Brazilian (Baependi Heart Study (BHS) [36] and Advento [37]) prospective cohort studies. SSSH participants completed a supplementary questionnaire (R/S survey) and cohorts provided blood samples and historic questionnaire data that was centrally harmonized at Massachusetts General Hospital. Each cohort study obtained written informed consent from their participants as well as institutional review board approval for cohort maintenance and participation in the SSSH. SSSH data coordination and analyses were approved by the Partners Human Research Committee. The current analysis includes 3,243 participants from five SSSH participating cohorts: BHS (N = 386), BWHS (N = 957), MASALA (n = 505), NHSII (N = 1,097), and SHS (N = 279).

**Cohort descriptions.** The Baependi Heart Study (BHS) is a family-based cohort in which residents from Baependi, a town in Minas Gerais–Brazil, were randomly selected from 11 out of 12 census districts, followed by all proband's extended family members who were older than 18 years old and willing to take part. Details on recruitment have been thoroughly described previously [36]. All participants provided informed written consent. The present analysis includes 386 BHS subjects aged 18 to 88 years old that provided a blood sample in 2005 or 2006 and completed the SSSH R/S survey between 2016–2019. We excluded 48 participants with missing information on sexual abuse and 31 missing data on R/S measures.

BWHS began in 1995 when 59,000 African American women aged 21–69 years were enrolled through self-administered questionnaires mailed to subscribers of *Essence* magazine, members of Black women's professional organizations as well as the friends and relatives of early respondents [31]. Information on demographics, lifestyle factors, and medical history is collected at baseline and updated biennially. Between 2016–2017, a random sample of 1000 BWHS participants who provided a blood sample between 2013–2015 were selected and invited to complete the R/S survey. Of 997 participants that successfully completed the telomere assay, we excluded participants with missing data on physical (N = 183) or sexual abuse (N = 32) leaving a final analytic sample of 814 BWHS participants for analyses of physical abuse and 965 for analyses of sexual abuse.

MASALA is a prospective, community-based cohort of 906 immigrant South Asian men and women, aged 40–84 years old, recruited from two clinical field centers (University of California, San Francisco and Northwestern University, Chicago) between October 2010 and March 2013 [33]. At their in-person baseline examination, MASALA participants completed questionnaires on sociodemographic, lifestyle and behavioral factors, and provided a fasting blood sample. Of the 906 participants, 696 provided consent for DNA extraction. Between September 2015 and March 2018, participants were invited back for a follow-up examination

(Exam 2) where the SSSH R/S survey was administered. Three participants failed the telomere assay, and we excluded participants with missing information on physical abuse (n = 166) or R/S measures (n = 22), leaving a final sample of 505. Information on sexual abuse was not available in MASALA.

NHSII began in 1989 with 116,430 female registered nurses between the ages of 25 and 42. Participants are invited to complete biennial questionnaires to assess sociodemographic, lifestyle, behavioral, and medical information. In 2016, participants were invited to complete a web-based version of the R/S survey. Eligibility criteria included: an email address, provision of a blood sample in at least two cohort blood collections, age 45–75 at the time of their most recent blood draw (in 2010–2013), and completion of four questionnaires (the 2001 violence questionnaire, the 2008 trauma questionnaire, and the 2013 and 2015 main questionnaires), and no active participation in an ongoing ancillary study. Of the 4,251participants that completed the R/S survey, 1129 had their most recent blood sample assayed for relative telomere length. We excluded participants with missing information on physical or sexual abuse (n = 2) or R/S measures (n = 30) for a final sample size of 1097.

SHS is a population-based longitudinal cohort of 4,549 American Indians from 13 communities in Arizona, North and South Dakota, and Oklahoma which began in 1989 [35]. The SHS has completed three clinical examinations of the original Cohort (Phase I: 1989–1991; Phase II: 1993–1995; Phase III: 1998–1999, respectively). In Phase IV, an additional 18 to 25 extended families (a total of about 900 members at least 15 years of age) were recruited from each of the field centers from 2001–2003. In 2006–2009, Phase V a second exam of the family cohort and continued surveillance of the original cohort was completed. Our analysis includes participants in Phase IV or Phase V, who completed the R/S survey (N = 709) and had available telomere assay data (N = 365) at the time this analysis. We excluded participants with missing family ID (n = 16), data on physical (N = 61) or sexual abuse (N = 62), or R/S (N = 9) leaving a final analytic sample of 279 SHS participants for analyses of physical abuse and 278 for analyses of sexual abuse.

## Physical abuse

Physical abuse was assessed in childhood (age <12) or adolescence (age 12–18) using questions from the Revised Conflict Tactics Scale (CTS2) [38] which asked participants to report how often an adult caregiver pushed, grabbed, or shoved; kicked, bit, or punched; hit with something that hurt; choked or burned; or physically attacked the participant. We assigned respondents 1 point for each report of a physical abuse item that occurred >4 times, except for "choked or burned" or "seriously harmed someone I loved" where we assigned 1 point if they occurred 1 to 3 times and 2 points if they occurred 4 times. The resulting scores were categorized using the following groupings: no abuse (0), mild abuse (1), moderate abuse (2), severe abuse (≥3) [39]. We evaluated the impact of physical abuse in childhood or adolescence separately, and cross-classified yes/no indicators of physical abuse in childhood and adolescence to generate a four-level variable that indicated no physical abuse, physical abuse in childhood only, adolescence only, or childhood and adolescence.

## Sexual abuse

In NHSII, sexual abuse in childhood (age <12) or adolescence (age 12–18) was assessed using questions from a 1995 Gallup Organization national telephone survey [40]. Participants were asked about forced sexual touching and forced sexual activity. Response categories were: "No, this never happened", "Yes, this happened once", or "Yes, this happened more than once". We categorized participants as no abuse (never touched and never forced sex), moderate abuse

(never touched or touched once and forced sex once; OR touched once or more and never forced sex), or severe abuse (forced sex more than once; OR touched more than once and forced sex once). In BHS, BWHS, and SHS, using questions from the CTS2, participants reported whether someone was "sexual with me against my will" or "exposed their genitals against my will". Response categories were never, 1 to 3 times, 4 or more times. We categorized participants as no abuse (never exposed genitals and never sexually abused), moderate abuse (exposed genitals or sexually abused 1–3 times), or severe abuse (exposed genitals or sexually abused ≥4 times). We evaluated the impact of sexual abuse in childhood or adolescence separately, and cross-classified yes/no indicators of sexual abuse in childhood and adolescence to generate a four-level variable that indicated no sexual abuse, sexual abuse in childhood only, adolescence only, or childhood and adolescence. We also cross classified physical and sexual abuse to create a five-level variable that ranged from no physical or sexual abuse to severe physical and severe sexual abuse. These abuse definitions have been used in previous publications [39, 41, 42].

## Religion and Spirituality (R/S)

Using data from the SSSH R/S survey, completed between 2016 and 2019, we classified participants into two groups, those who reported being "very religious or spiritual" compared to "moderately", "slightly" or "not at all". Positive and negative religious coping was assessed using items from the Religious Coping Scale [43]. Participants were asked, "When dealing with recent stressful situations. . ." to what extent did they cope using any of seven positive attributes (e.g., "I saw my situation as part of God's plan", "I sought God's love or care,") or six negative attributes (e.g., "I wondered what I did for God to punish me," "I questioned [God]'s love or care for me"). Items were scored from 1 (not at all) to 4 (a great deal), summed, and averaged to create a single continuous variable. Participants in each cohort were stratified by positive and negative religious coping scores using study-specific medians.

## Relative leukocyte telomere length (RTL)

Blood collection and processing methods for each cohort have been described previously [33, 44–47]. Genomic DNA was extracted from peripheral blood leukocytes using the QIAmp (Qiagen) 96-spin blood protocol (BWHS and NHSII), phenol-chloroform standard protocol (SHS and BHS), or sodium dodecylsulfate cell lysis followed by a salt precipitation (MASALA). RTL assays for BWHS, MASALA, NHSII, and SHS samples were performed in the laboratory of Dr. Immaculata De Vivo (Boston, MA, USA), while BHS samples were assayed at the Laboratory of Genetics and Molecular Cardiology, Heart Institute (Incor) (São Paulo, Brazil). Both laboratories assayed each sample in triplicate using quantitative real-time polymerase chain reaction (qPCR) according to Cawthon et al.'s protocol [48]. RTL was calculated as the exponentiated ratio of telomere repeat copy number to single-copy gene (36B4) copy number (T/S) corrected for a reference sample [48]. RTL is reported as the exponentiated T/S ratio. RTL correlates with absolute telomere lengths determined by Southern blot (r = 0.68; p<0.001).[48] The total intra assay coefficients of variation (CV) ranged from 0.27% (NHSII) to 0.35% (MASALA) and the inter assay CVs ranged from 0.33% (MASALA) to 0.77% (SHS). CVs for the exponentiated T:S ratio ranged from 4.7% (SHS) to 10.3% (NHS). We excluded extreme outliers (>3 interquartiles above the 75th percentile value and < 3 interquartiles below the 25th percentile). To minimize the impact of potential batch effects on RTL measurements across different studies, we calculated z scores of log-transformed RTL by standardizing the value in comparison with the mean within each individual study-batch [49].

## Statistical analysis

We used linear regression with generalized estimating equations (GEE) and an independence working correlation, to generate robust standard errors which accounts for relatedness among family members (in BHS and SHS). We modeled the association between severity of physical, sexual, or joint physical and sexual abuse experienced in childhood and/or adolescence and RTL z-scores (hereafter called RTL). Since RTL z scores are challenging to interpret, we converted beta coefficients ($\beta$) for point estimates and 95% confidence intervals into unitless relative differences (% differences) using the formula ($e^{\beta} - 1$)$X$ 100 [50].

We present three models, first adjusted for age at blood draw only, next adding demographic and childhood factors (race/ethnicity, gender, parental educational attainment at participant's birth or during childhood, parental home ownership at participant's birth or during childhood, loss of parent before age 18, childhood financial hardship, and receipt of public assistance in childhood), and lastly, to examine to what extent any observed associations are driven by lifestyle and behavioral factors, we adjusted for potential adult mediators including body mass index (BMI), household income, smoking status, physical activity, alternative healthy eating index (AHEI), and depressive symptoms. Childhood financial hardship and receipt of public assistance were not available in BHS or NHSII and AHEI was not available in BHS.

We conducted cohort-specific analyses and pooled estimates using the random effects model by DerSimonian and Laird which takes into account within and between study variation (heterogeneity) [51]. Between-study heterogeneity was evaluated with Q statistics [52]. To investigate potential sources of heterogeneity, we conducted subgroup analyses restricted to females [53] and restricted NHSII to white participants due to known differences in telomere length between populations of European and African ancestry [54].

We assessed potential effect modification by R/S variables using a cross-product term of ordinal sexual abuse and dichotomous R/S variables. We present stratified results along with the Wald p-value for the interaction term. To estimate p for linear trend we modeled the ordinal abuse exposure variable as continuous. All P values are two sided, and an α level of 0.05 was used. We used SAS version 9.4 (SAS Institute, Cary, NC) and R version 3.6.0 (R Foundation for Statistical Computing, Vienna, Austria) for all statistical analyses.

## Results

Participant characteristics by cohort and severity of physical and sexual abuse in childhood are displayed in Tables 1 and 2 respectively. Compared to no abuse, severe physical abuse in childhood was associated with childhood financial hardship, greater physical activity (except in SHS). Reported sexual abuse did not differ by most childhood characteristics. The prevalence of reported physical abuse in childhood and adolescence was 31.6% overall (range: 15.3% in BHS to 64.7% in BWHS) and was 35.0% for sexual abuse (range: 3.6% in BHS to 50.4% in BWHS).

In our pooled analysis, severe physical abuse was not associated with RTL, although we did observe an association for moderate physical abuse in childhood (Table 3). In models adjusted for childhood and demographic factors, we observed no association between severe physical abuse and telomere length in childhood (percent difference (PD): 2.0%, 95% CI: -9.5, 15.0; p-trend = 0.66) or adolescence (p-trend = 0.05). Compared to no abuse, moderate physical abuse in childhood was associated with 18.9% lower RTL (95% CI: -31.6%, -3.9%; p-heterogeneity = 0.89) after adjustment for childhood and demographic covariates (model 2). Results were unchanged by adjustment for adult factors.

**Table 1. Participant characteristics according to experience of physical abuse in childhood (Age 0–11) and cohort.**

| Variable | BHS (N = 386) | | | | BWHS (N = 814) | | | | MASALA (N = 505) | | | | NHSII (N = 1097) | | | | SHS (N = 279) | | | |
|---|---|---|---|---|---|---|---|---|---|---|---|---|---|---|---|---|---|---|---|---|
| | None | Mild | Moderate | Severe | None | Mild | Moderate | Severe | None | Mild | Moderate | Severe | None | Mild | Moderate | Severe | None | Mild | Moderate | Severe |
| n(%)[1] | 329 (85.2) | 27 (7.0) | 12 (3.1) | 18 (4.7) | 331 (40.7) | 153 (18.8) | 83 (10.2) | 247 (30.3) | 437 (86.5) | 44 (8.7) | 18 (3.6) | 6 (1.2) | 923 (84.1) | 111 (10.1) | 41 (3.7) | 22 (2.0) | 176 (63.1) | 34 (12.2) | 13 (4.7) | 56 (20.1) |
| Log-RTL (mean (SD))[2] | -0.04 (1.0) | 0.07 (1.1) | -0.48 (0.9) | -0.12 (1.0) | 0.0 (1.0) | -0.1 (1.0) | -0.1 (1.1) | 0.0 (0.9) | 0.0 (1.0) | 0.1 (0.9) | -0.1 (1.0) | -0.2 (0.6) | 0.0 (1.0) | 0.1 (0.9) | -0.5 (1.1) | 0.5 (0.7) | 0.1 (1.0) | 0.5 (0.8) | 0.1 (0.4) | 0.2 (0.8) |
| Age at blood draw (yrs) (mean (SD)) | 46.8 (15.8) | 47.1 (16.0) | 47.5 (17.1) | 48.9 (15.7) | 56.9 (7.5) | 55.7 (7.4) | 55.9 (7.7) | 55.9 (7.6) | 55.4 (9.3) | 53.5 (8.4) | 51.4 (9.4) | 53.3 (11.8) | 57.3 (4.3) | 57.2 (4.4) | 57.7 (4.1) | 58.4 (4.8) | 40.6 (13.6) | 41.1 (12.7) | 38.9 (8.4) | 38.7 (11.2) |
| Positive coping, (mean (SD)) | 3.9 (0.3) | 3.9 (0.3) | 3.8 (0.4) | 3.9 (0.2) | 3.2 (0.7) | 3.2 (0.7) | 3.1 (0.8) | 3.1 (0.8) | 2.6 (0.9) | 2.7 (0.9) | 2.2 (0.9) | 3.2 (0.8) | 2.8 (0.9) | 2.8 (1.0) | 2.7 (1.0) | 2.8 (0.7) | 2.8 (0.8) | 2.8 (0.6) | 2.4 (0.5) | 2.5 (0.8) |
| Negative coping, (mean (SD)) | 1.9 (0.7) | 2.1 (0.8) | 2 (0.7) | 2.1 (0.8) | 1.5 (0.6) | 1.4 (0.6) | 1.4 (0.4) | 1.4 (0.5) | 1.4 (0.6) | 1.4 (0.6) | 1.5 (0.4) | 1.4 (0.3) | 1.2 (0.4) | 1.3 (0.4) | 1.2 (0.3) | 1.3 (0.4) | 1.6 (0.6) | 1.8 (0.5) | 1.6 (0.3) | 1.6 (0.6) |
| AHEI, (mean (SD))[3] | NA | NA | NA | NA | 41.8 (10.7) | 43.1 (10.6) | 42.3 (11.4) | 41.8 (10.4) | 70.3 (6.5) | 68.7 (6.3) | 69.7 (5.3) | 73.0 (3.6) | 66.3 (12.8) | 64.8 (12.6) | 66.3 (12.1) | 68.3 (10.7) | 44.5 (8.0) | 42.8 (6.5) | 46.4 (4.6) | 43.7 (6.7) |
| Physical activity, (mean (SD))[4] | NA | NA | NA | NA | 9.3 (6.7) | 11.1 (11.2) | 10.3 (7.7) | 12.2 (18.7) | 22.9 (21.2) | 18.8 (17.9) | 25.9 (17.0) | 39.5 (26.5) | 25.8 (26.8) | 28.7 (41.0) | 31.7 (25.2) | 39.9 (26.1) | 5786 (3509) | 5938 (4328) | 6767 (1960) | 6728 (2855) |
| Very religious or spiritual, n (%) | NA | NA | NA | NA | 144 (43) | 71 (47) | 37 (49) | 116 (46) | 74 (17) | 11 (30) | 2 (6) | 2 (29) | 385 (42) | 49 (45) | 17 (37) | 6 (24) | 51 (30) | 10 (25) | 2 (19) | 16 (27) |
| Female, n (%) | 204 (62) | 22 (81.5) | 9 (75) | 11 (61.1) | 331 (100) | 153 (100) | 83 (100) | 247 (100) | 187 (43) | 11 (26) | 4 (24) | 2 (34) | 923 (100) | 111 (100) | 41 (100) | 22 (100) | 118 (68) | 25 (73) | 8 (63) | 24 (36) |
| **Race, n(%)** | | | | | | | | | | | | | | | | | | | | |
| White | 218 (66.3) | 13 (48.1) | 6 (50) | 11 (61.1) | 0 (0) | 0 (0) | 0 (0) | 0 (0) | 0 (0) | 0 (0) | 0 (0) | 0 (0) | 875 (99) | 102 (96) | 39 (98) | 22 (100) | 4 (2) | 0 (0) | 0 (0) | 1 (0) |
| Black | 14 (4.3) | 3 (11.1) | 0 (0) | 2 (11.1) | 331 (100) | 153 (100) | 83 (100) | 247 (100) | 0 (0) | 0 (0) | 0 (0) | 0 (0) | 1 (0) | 3 (3) | 0 (0) | 0 (0) | 0 (0) | 0 (0) | 0 (0) | 0 (0) |
| Mulatto/ Brown | 84 (25.6) | 11 (40.7) | 6 (50) | 5 (27.8) | NA | NA | NA | NA | NA | NA | NA | NA | NA | NA | NA | NA | NA | NA | NA | NA |
| American Indian | 0 (0) | 0 (0) | 0 (0) | 0 (0) | 0 (0) | 0 (0) | 0 (0) | 0 (0) | 0 (0) | 0 (0) | 0 (0) | 0 (0) | 5 (1) | 1 (1) | 0 (0) | 0 (0) | 170 (97) | 34 (100) | 13 (100) | 55 (100) |
| South Asian | 0 (0) | 0 (0) | 0 (0) | 0 (0) | 0 (0) | 0 (0) | 0 (0) | 0 (0) | 437 (100) | 44 (100) | 18 (100) | 6 (100) | 3 (0) | 1 (1) | 1 (2) | 0 (0) | 0 (0) | 0 (0) | 0 (0) | 0 (0) |
| Other | 4 (1.2) | 0 (0) | 0 (0) | NA | 0 (0) | 0 (0) | 0 (0) | 0 (0) | 0 (0) | 0 (0) | 0 (0) | 0 (0) | 0 (0) | 0 (0) | 0 (0) | 0 (0) | 0 (0) | 0 (0) | 0 (0) | 0 (0) |
| **Parental home ownership, n(%)** | NA | NA | NA | NA | 182 (58) | 80 (54) | 51 (66) | 128 (52) | 272 (63) | 24 (58) | 89 (38) | 5 (76) | 469 (54) | 53 (50) | 17 (47) | 9 (47) | 106 (60) | 16 (44) | 8 (55) | 24 (44) |

*(Continued)*

Table 1. (Continued)

| Variable | BHS (N = 386) | | | | BWHS (N = 814) | | | | MASALA (N = 505) | | | | NHSII (N = 1097) | | | | SHS (N = 279) | | | |
|---|---|---|---|---|---|---|---|---|---|---|---|---|---|---|---|---|---|---|---|---|
| | None | Mild | Moderate | Severe | None | Mild | Moderate | Severe | None | Mild | Moderate | Severe | None | Mild | Moderate | Severe | None | Mild | Moderate | Severe |
| **Childhood financial hardship, n (%)** | NA | NA | NA | NA | 63 (21) | 41 (26) | 18 (22) | 97 (42) | 116 (28) | 9 (32) | 3 (8) | 1 (28) | NA | NA | NA | NA | 65 (48) | 13 (44) | 8 (71) | 21 (45) |
| **Received public assistance, (%)** | NA | NA | NA | NA | 67 (21) | 34 (21) | 26 (30) | 86 (38) | 7 (2) | 3 (10) | 1 (4) | 0 (0) | NA | NA | NA | NA | 38 (38) | 9 (66) | 2 (72) | 9 (57) |
| **Mother's education level, n(%)** | | | | | NA | NA | NA | NA | | | | | | | | | | | | |
| <12 grade | 303 (92.1) | 25 (92.6) | 11 (91.7) | 18 (100) | 97 (30) | 36 (25) | 18 (27) | 64 (27) | 173 (39) | 19 (53) | 7 (39) | 2 (24) | 70 (8) | 12 (11) | 7 (14) | 1 (3) | 47 (27) | 10 (27) | 4 (39) | 14 (30) |
| high school degree or GED | 20 (6.1) | 1 (3.7) | 1 (8.3) | 0 (0) | 111 (33) | 50 (33) | 29 (31) | 69 (29) | 127 (30) | 13 (27) | 1 (4) | 2 (38) | 485 (56) | 62 (58) | 20 (57) | 14 (70) | 68 (43) | 8 (26) | 5 (30) | 20 (34) |
| some college or vocational school | 5 (1.5) | 1 (3.7) | 0 (0) | 0 (0) | 65 (21) | 42 (29) | 23 (32) | 64 (27) | 41 (10) | 2 (3) | 1 (8) | 0 (0) | 207 (24) | 21 (22) | 10 (24) | 5 (25) | 24 (14) | 8 (31) | 2 (14) | 8 (15) |
| college graduate or higher | 1 (0.3) | 0 (0) | 0 (0) | 0 (0) | 55 (18) | 22 (14) | 12 (10) | 44 (17) | 91 (22) | 10 (17) | 9 (50) | 2 (38) | 104 (12) | 8 (9) | 3 (5) | 1 (2) | 23 (15) | 4 (16) | 2 (17) | 10 (21) |
| **Father's education level, n(%)** | | | | | | | | | | | | | | | | | | | | |
| <12 grade | 313 (95.1) | 27 (100) | 12 (100) | 17 (94.4) | 109 (35) | 54 (40) | 27 (36) | 77 (34) | 64 (15) | 6 (13) | 1 (10) | 1 (4) | 99 (12) | 14 (13) | 8 (22) | 2 (5) | 58 (41) | 9 (22) | 2 (28) | 17 (28) |
| high school degree or GED | 12 (3.6) | 0 (0) | 0 (0) | 0 (0) | 92 (30) | 44 (31) | 19 (23) | 69 (29) | 98 (22) | 9 (18) | 2 (9) | 1 (14) | 416 (48) | 50 (48) | 15 (45) | 11 (53) | 57 (40) | 14 (56) | 7 (61) | 19 (45) |
| some college or vocational school | 4 (1.2) | 0 (0) | 0 (0) | 1 (5.6) | 55 (19) | 24 (16) | 21 (26) | 45 (20) | 46 (11) | 6 (12) | 3 (19) | 1 (24) | 140 (16) | 19 (19) | 9 (19) | 1 (3) | 20 (14) | 2 (5) | 1 (11) | 8 (14) |
| college graduate or higher | 0 (0) | 0 (0) | 0 (0) | 0 (0) | 45 (16) | 17 (13) | 13 (15) | 41 (16) | 225 (52) | 23 (56) | 12 (62) | 3 (58) | 202 (24) | 21 (20) | 8 (14) | 6 (38) | 7 (5) | 5 (16) | 0 (0) | 4 (13) |
| **Loss of mother before age 18, n(%)** | 17 (2%) | 3 (3%) | 2 (4%) | 2 (5%) | 10 (3) | 5 (3) | 5 (5) | 14 (5) | 14 (3) | 2 (8) | 1 (6) | 1 (20) | 17 (2) | 3 (3) | 2 (4) | 2 (5) | 10 (6) | 2 (3) | 1 (11) | 5 (7) |
| **Loss of father before age 18, n(%)** | 65 (7%) | 8 (9%) | 4 (10%) | 4 (11%) | 48 (15) | 31 (21) | 11 (13) | 51 (21) | 31 (7) | 7 (23) | 0 (0) | 1 (14) | 65 (7) | 8 (9) | 4 (10) | 4 (11) | 31 (18) | 4 (10) | 4 (36) | 13 (20) |

(Continued)

Table 1. (Continued)

| Variable | BHS (N = 386) | | | | BWHS (N = 814) | | | | MASALA (N = 505) | | | | NHSII (N = 1097) | | | | SHS (N = 279) | | | |
|---|---|---|---|---|---|---|---|---|---|---|---|---|---|---|---|---|---|---|---|---|
| | None | Mild | Moderate | Severe | None | Mild | Moderate | Severe | None | Mild | Moderate | Severe | None | Mild | Moderate | Severe | None | Mild | Moderate | Severe |
| **Adult BMI, n, %** | | | | | | | | | | | | | | | | | | | | |
| <25 | 137 (41.6) | 8 (29.6) | 5 (41.7) | 5 (27.8) | 93 (28) | 36 (22) | 21 (31) | 74 (32) | 197 (45) | 19 (42) | 7 (39) | 2 (44) | 456 (53) | 63 (61) | 17 (45) | 12 (56) | 40 (22) | 7 (21) | 5 (26) | 9 (18) |
| 25–29.9 | 101 (30.7) | 11 (40.7) | 4 (33.3) | 7 (38.9) | 99 (30) | 55 (36) | 21 (29) | 70 (30) | 183 (42) | 21 (51) | 10 (52) | 4 (56) | 247 (28) | 26 (23) | 8 (21) | 3 (22) | 53 (29) | 10 (25) | 3 (27) | 12 (23) |
| ≥ 30 | 69 (21.0) | 7 (25.9) | 1 (8.3) | 6 (33.3) | 137 (41) | 62 (42) | 41 (39) | 101 (39) | 54 (12) | 4 (7) | 1 (9) | 0 (0) | 165 (19) | 19 (16) | 13 (34) | 4 (23) | 82 (48) | 17 (53) | 5 (47) | 35 (59) |
| **Adult Smoking Status, n (%)** | | | | | | | | | | | | | | | | | | | | |
| Never smoker | 218 (66.3) | 17 (63.0) | 7 (58.3) | 10 (55.6) | 246 (76) | 11 (76) | 61 (74) | 172 (70) | 364 (83) | 34 (78) | 14 (81) | 4 (62) | 647 (70) | 70 (63) | 22 (55) | 9 (51) | 65 (35) | 10 (36) | 4 (34) | 16 (31) |
| Former smoker | 79 (24.0) | 6 (22.2) | 2 (16.7) | 7 (38.9) | 61 (17) | 24 (17) | 20 (23) | 52 (21) | 61 (14) | 7 (13) | 3 (14) | 2 (38) | 224 (25) | 37 (34) | 17 (35) | 12 (47) | 40 (24) | 9 (18) | 1 (3) | 11 (16) |
| Current smoker | 25 (7.6) | 4 (14.8) | 3 (25.0) | 1 (5.6) | 24 (7) | 10 (7) | 2 (2) | 23 (9) | 12 (3) | 3 (8) | 1 (4) | 0 (0) | 52 (6) | 4 (3) | 2 (11) | 1 (2) | 71 (41) | 15 (46) | 8 (64) | 29 (54) |
| **Adult Depressive Symptoms, n(%)** | 3 (0.9) | 0 (0) | 0 (0) | 0 (0) | 60 (18) | 32 (21) | 16 (20) | 66 (26) | 40 (9) | 11 (30) | 2 (11) | 0 (0) | 49 (6) | 6 (5) | 2 (4) | 2 (5) | 27 (30) | 14 (36) | 1 (35) | 12 (27) |

[1] Percentage presented are of the total sample in each cohort. All other percentages in table are column percentages.

[2] Log-transformed relative telomere length.

[3] Alternative Healthy Eating Index.

[4] Value is MET-hours per week in NHSII, BWHS, and MASALA, and steps per week in SHS.

NA—Not avaialble or Not applicable.

**Table 2. Participant characteristics according to experience of sexual abuse in childhood (Age 0–11) and cohort.**

| Variable | BHS (N = 352) | | | BWHS(N = 965) | | | NHSII (N = 1097) | | | SHS(N = 278) | | |
|---|---|---|---|---|---|---|---|---|---|---|---|---|
| | None | Moderate | Severe | None | Moderate | Severe | None | Moderate | Severe | None | Moderate | Severe |
| n(%)[1] | 342 (97.2) | 9(2.6) | 1(0.28) | 626 (64.9) | 221(22.9) | 118 (12.2) | 842 (76.8) | 128(11.7) | 127 (11.6) | 242(87.1) | 23(8.3) | 13(4.7) |
| Log-RTL (mean (SD))[2] | -0.02 (1.0) | 0.28(0.7) | -1.66 (NA) | 0.0(1.0) | 0.1(1.0) | -0.1(1.1) | 0.0(1.0) | 0.1(0.9) | -0.2(1.0) | 0.1(1.0) | 0.3(0.6) | 0.0(0.5) |
| Age at blood draw (yrs) (mean (SD)) | 47.0 (15.8) | 46.1(10.7) | 57(NA) | 56.2 (7.5) | 56.5(7.5) | 56.6(7.5) | 57.4 (4.4) | 57.4(4.3) | 57.1(4.0) | 40.3 (13.0) | 38.0(11.6) | 40.4(9.7) |
| Positive coping, (mean (SD)) | 3.9(0.3) | 3.9(0.1) | 4 (NA) | 3.1(0.8) | 3.1(0.8) | 3.2(0.7) | 2.8(0.9) | 2.7(1.0) | 2.8(0.9) | 2.7(0.8) | 3.0(0.7) | 2.5(0.6) |
| Negative coping, (mean (SD)) | 1.9(0.7) | 1.7(0.7) | 1.7(NA) | 1.5(0.5) | 1.4(0.4) | 1.4(0.4) | 1.2(0.4) | 1.2(0.3) | 1.2(0.3) | 1.7(0.6) | 2.0(0.6) | 1.6(0.3) |
| AHEI, (mean (SD))[3] | NA | NA | NA | 42.1 (10.4) | 42.5(10.2) | 43.5 (10.8) | 66.2 (12.8) | 66.6(12.6) | 63.7 (13.2) | 44.2(8.1) | 43.7(5.0) | 48.3(6.1) |
| Physical activity, (mean (SD))[4] | NA | NA | NA | 10.4 (8.9) | 11.5(10.8) | 11.7 (24.2) | 26.9 (30.7) | 23.8(26.6) | 27.3 (24.6) | 6012 (3492) | 6905 (4208) | 4846 (1486) |
| Very religious or spiritual, n (%) | NA | NA | NA | 275(44) | 98(44) | 67(59) | 366(43) | 44(34) | 47(38) | 176(27) | 15(43) | 9(28) |
| Female, n(%) | 214 (62.6) | 8(88.9) | 1(100) | 626 (100) | 221(100) | 118(100) | 842 (100) | 128(100) | 127(100) | 146(61) | 18(81) | 10(67) |
| Race, n(%) | | | | | | | | | | | | |
| White | 226 (66.1) | 6(66.7) | 0(0) | 0(0) | 0(0) | 0(0) | 802(99) | 117(97) | 119(96) | 5(2) | 0(0) | 0(0) |
| Black | 17(5.0) | 0(0) | 0(0) | 626 (100) | 221(100) | 118(100) | 0(0) | 1(1) | 3(2) | 0(0) | 0(0) | 0(0) |
| Mulatto/Brown | 94(27.5) | 3(33.3) | 1(100) | NA | NA | NA | NA | NA | NA | NA | NA | NA |
| American Indian | 0(0) | 0(0) | 0(0) | 0(0) | 0(0) | 0(0) | 4(1) | 1(1) | 1(2) | 235(97) | 23(100) | 13(100) |
| South Asian | 0(0) | 0(0) | 0(0) | 0(0) | 0(0) | 0(0) | 4(0) | 1(1) | 0(0) | 0(0) | 0(0) | 0(0) |
| Other | 4(1.2) | 0(0) | 0(0) | 0(0) | 0(0) | 0(0) | 0(0) | 0(0) | 0(0) | 0(0) | 0(0) | 0(0) |
| Parental home ownership, n (%) | NA | NA | NA | 339(55) | 106(50) | 55(45) | 431(54) | 62(55) | 55(45) | 138(57) | 10(42) | 5(48) |
| Childhood financial hardship, n(%) | NA | NA | NA | 145(25) | 80(37) | 39(38) | NA | NA | NA | 85(46) | 12(51) | 9(80) |
| Received public assistance, (%) | NA | NA | NA | 141(24) | 75(35) | 40(33) | NA | NA | NA | 79(43) | 14(65) | 9(80) |
| Mother's education level, n (%) | | | | | | | | | | | | |
| <12 grade | 314 (91.2) | 8(88.9) | 2(0) | 158(26) | 67(30) | 28(22) | 69(9) | 9(8) | 12(10) | 64(28) | 5(17) | 5(46) |
| high school degree or GED | 21(6.1) | 1(11.1) | 1(100) | 207(33) | 63(30) | 37(29) | 435(55) | 70(60) | 76(64) | 90(41) | 7(33) | 4(31) |
| some college or vocational school | 6(1.8) | 0(0) | 0(0) | 143(23) | 51(24) | 30(30) | 193(24) | 26(22) | 24(22) | 38(17) | 4(20) | 0(0) |
| college graduate or higher | 1(0.3) | 0(0) | 0(0) | 108(17) | 36(16) | 20(18) | 98(12) | 13(10) | 5(4) | 30(13) | 6(30) | 3(23) |
| Father's education level, n (%) | | | | | | | | | | | | |
| <12 grade | 325 (95.0) | 9(100) | 1(100) | 207(36) | 69(34) | 37(34) | 93(12) | 14(12) | 16(14) | 76(38) | 5(23) | 4(34) |
| high school degree or GED | 12(3.5) | 0(0) | 0(0) | 170(29) | 63(31) | 33(29) | 364(46) | 64(55) | 64(57) | 81(41) | 10(55) | 6(58) |
| some college or vocational school | 5(1.5) | 0(0) | 0(0) | 114(20) | 41(21) | 21(21) | 136(17) | 16(14) | 17(14) | 28(14) | 2(16) | 1(8) |
| college graduate or higher | 0(0) | 0(0) | 0(0) | 90(19) | 30(15) | 15(15) | 196(25) | 23(18) | 18(16) | 14(7) | 2(6) | 0(0) |
| Loss of mother before age 18, n(%) | NA | NA | NA | 23(4) | 14(6) | 5(4) | 15(2) | 3(3) | 6(4) | 16(7) | 1(2) | 1(7) |
| Loss of father before age 18, n(%) | NA | NA | NA | 100(16) | 47(22) | 23(17) | 61(7) | 8(6) | 12(11) | 49(21) | 2(5) | 1(7) |

*(Continued)*

**Table 2.** (Continued)

| | BHS (N = 352) | | | BWHS(N = 965) | | | NHSII (N = 1097) | | | SHS(N = 278) | | |
|---|---|---|---|---|---|---|---|---|---|---|---|---|
| Variable | None | Moderate | Severe | None | Moderate | Severe | None | Moderate | Severe | None | Moderate | Severe |
| **Adult BMI, n, %** | | | | | | | | | | | | |
| <25 | 144 (42.1) | 1(11.1) | 0(0) | 177(28) | 58(26) | 23(21) | 434(55) | 63(50) | 51(43) | 55(23) | 5(20) | 1(8) |
| 25–29.9 | 104 (30.4) | 3(33.3) | 1(100) | 184(30) | 69(30) | 34(32) | 211(27) | 34(28) | 39(34) | 70(29) | 7(33) | 1(7) |
| ≥ 30 | 74(21.6) | 5(55.5) | 0(0) | 262(42) | 93(44) | 60(47) | 142(18) | 29(22) | 30(23) | 116(48) | 11(47) | 11(86) |
| **Adult Smoking Status, n(%)** | | | | | | | | | | | | |
| Never smoker | 223 (65.2) | 5(55.6) | 0(0) | 472(75) | 148(66) | 83(72) | 592(70) | 79(59) | 77(59) | 88(36) | 5(24) | 2(15) |
| Former smoker | 89(26.0) | 2(22.2) | 0(0) | 108(17) | 54(25) | 27(21) | 206(24) | 42(36) | 42(35) | 52(22) | 6(30) | 2(11) |
| Current smoker | 28(8.2) | 2(22.2) | 1(100) | 46(7) | 19(8) | 8(6) | 44(5) | 7(5) | 8(6) | 102(42) | 12(46) | 9(74) |
| **Adult Depressive Symptoms, n(%)** | 2(0.6) | 0(0) | 0(0) | 112(18) | 56(25) | 36(28) | 46(6) | 7(6) | 6(7) | 55(27) | 12(49) | 6(53) |

[1] Percentage presented are of the total sample in each cohort. All other percentages in table are column percentages.

[2] Log-transformed relative telomere length.

[3] Alternative Healthy Eating Index.

[4] Value is MET-hours per week in NHSII, BWHS, and MASALA, and steps per week in SHS.

NA—Not avaialble or Not applicable.

Severe sexual abuse in childhood or adolescence was associated with decreased RTL in our pooled analysis, as was any sexual abuse experienced in childhood and adolescence (Table 4). Compared to no abuse, severe sexual abuse in childhood was associated with 13.9% lower RTL (95% CI: -23.7%, -2.0%; p-trend = 0.15; p-heterogeneity = 0.56) in age-adjusted models, and 15.6% lower after adjustment for childhood and demographic covariates (95% CI: -25.9%, -4.9%; p-trend = 0.04; p-heterogeneity = 0.58). Additional adjustment for adult covariates did not attenuate the association. Compared to no abuse, participants that reported severe sexual abuse in adolescence had 15.6% lower RTL (95% CI: -27.4%, -2.0%; p-trend = 0.18; p-heterogeneity = 0.80) in model 1, and 16.5% lower (95% CI: -28.1%, -3.0%; p-trend = 0.08; p-heterogeneity = 0.68) in model 2. Results were similar after accounting for adult factors in model 3. Sexual abuse experienced in both childhood and adolescence was associated with 11.3% lower RTL in model 2 (95% CI: -20.5%, -2.0%; p-trend = 0.03; p-heterogeneity = 0.62), and this association was maintained in model 3.

Our cross-classified measure of physical and sexual abuse was not associated with telomere length in the pooled analysis (Table 5). Compared to no abuse, severe physical and sexual abuse in childhood (-1.0%, 95% CI: -28.1%, 36.3%; p-trend = 0.15; p-heterogeneity = 0.06) or adolescence (-4.9%, 95% CI: -47.8%, 73.3%; p-trend = 0.06; p-heterogeneity = 0.07) was not associated with RTL. We did not observe evidence that the impact of physical and sexual abuse on RTL differed by the extent of religiosity/spirituality or level of positive or negative religious coping in our pooled or cohort-specific analyses (Table 6). In the pooled analysis, all p-values for the interaction term were >0.20.

## Discussion

In this study of 3,232 participants from five diverse prospective cohorts in the US and Brazil, we found that severe sexual abuse in childhood and/or adolescence was associated with shorter telomeres. 28.7% of participants reported any physical abuse and 23.8% reported any sexual

**Table 3. Severity of physical abuse in childhood or adolescence and adult telomere length.**

| Childhood (Age <12) | Model | BHS N = 386 n | BHS % Difference (95% CI) | BWHS N = 814 n | BWHS % Difference (95% CI) | MASALA N = 505 n | MASALA % Difference (95% CI) | NHSII N = 1097 n | NHSII % Difference (95% CI) | SHS N = 279 n | SHS % Difference (95% CI) | Pooled N = 3081 n | Pooled % Difference (95% CI) | p-het |
|---|---|---|---|---|---|---|---|---|---|---|---|---|---|---|
| None | Model 1ck[1] | 329 | ref | 331 | ref | 437 | ref | 923 | ref | 176 | ref | 2196 | ref | |
| Mild | | 27 | 11.6 (-24.5, 65.7) | 153 | -10.9 (-25.8, 7.0) | 44 | -4.7 (-29.7, 29.3) | 111 | 6.7 (-10.7, 27.6) | 34 | 35.8 (0.8, 82.9) | 369 | 4.1 (-8.6, 18.5) | 0.22 |
| Moderate | | 12 | -33.4 (-62.9, 19.4) | 83 | -13.0 (-32.9, 12.7) | 18 | -21.6 (-55.4, 38.0) | 41 | -26.8 (-47.2, 1.6) | 13 | -3.2 (-32.2, 38.2) | 167 | **-17.3 (-29.5, -2.0)** | 0.71 |
| Severe | | 18 | -6.8 (-42.5, 51.3) | 247 | -0.8 (-15.2, 16.1) | 6 | -10.4 (-66.6, 140.1) | 22 | 48.5 (-0.3, 121.2) | 56 | -7.8 (-25.7, 14.5) | 349 | -1.0 (-12.2, 15.0) | 0.34 |
| p-trend | | | 0.53 | | 0.86 | | 0.45 | | 0.73 | | 0.57 | | 0.58 | |
| None | Model 2[2] | 329 | ref | 331 | ref | 437 | ref | 923 | ref | 176 | ref | 2196 | ref | |
| Mild | | 27 | 9.7 (-26.2, 63.2) | 153 | -9.6 (-24.6, 8.3) | 44 | -2.0 (-26.6, 30.7) | 111 | 8.0 (-9.5, 28.9) | 34 | 21.3 (-2.5, 50.8) | 369 | 4.1 (-5.8, 16.2) | 0.38 |
| Moderate | | 12 | -34.0 (-63.3, 18.5) | 83 | -14.8 (-34.3, 10.5) | 18 | -18.4 (-51.8, 37.9) | 41 | -25.6 (-46.7, 3.8) | 13 | -12.4 (-41.2, 30.5) | 167 | **-18.9 (-31.6, -3.9)** | 0.89 |
| Severe | | 18 | -5.7 (-41.9, 53.0) | 247 | -1.4 (-15.9, 15.5) | 6 | -5.4 (-64.6, 152.9) | 22 | 44.6 (-4.1, 118.0) | 56 | -1.7 (-22.5, 24.6) | 349 | 2.0 (-9.5, 15.0) | 0.54 |
| p-trend | | | 0.52 | | 0.76 | | 0.56 | | 0.71 | | 0.82 | | 0.66 | |
| None | Model 3[3] | 329 | ref | 331 | ref | 437 | ref | 923 | ref | 176 | ref | 2196 | ref | |
| Mild | | 27 | 7.9 (-27.3, 60.2) | 153 | -10.0 (-24.8, 7.8) | 44 | -0.9 (-26.4, 33.6) | 111 | 7.6 (-9.6, 28.1) | 34 | 21.1 (0.7, 45.5) | 369 | 5.1 (-5.8, 18.5) | 0.26 |
| Moderate | | 12 | -37.1 (-64.9, 12.9) | 83 | -13.7 (-33.5, 11.9) | 18 | -18.7 (-51.3, 35.6) | 41 | -25.5 (-45.8, 2.5) | 13 | -10.5 (-43.1, 40.7) | 167 | **-18.9 (-31.6, -3.9)** | 0.82 |
| Severe | | 18 | 3.4 (-36.7, 68.7) | 247 | -0.8 (-15.5, 16.4) | 6 | -0.5 (-65.0, 182.7) | 22 | 39.6 (-6.7, 108.9) | 56 | -4.9 (-20.8, 14.2) | 349 | 1.0 (-10.4, 12.7) | 0.56 |
| p-trend | | | 0.66 | | 0.82 | | 0.62 | | 0.80 | | 0.61 | | 0.63 | |
| **Adolescence (Age 12–18)** | | n | % Difference (95% CI) | n | % Difference (95% CI) | n | % Difference (95% CI) | n | % Difference (95% CI) | n | % Difference (95% CI) | n | % Difference (95% CI) | |
| None | Model 1[1] | 347 | ref | 494 | ref | 466 | ref | 1037 | ref | 182 | ref | 2526 | ref | |
| Mild | | 19 | -10.7 (-43.5, 41.9) | 147 | 0.3 (-15.5, 19.0) | 30 | -8.3 (-41.2, 43.1) | 33 | -23.0 (-46.9, 11.5) | 41 | 3.5 (-13.7, 24.2) | 270 | -2.0 (-12.2, 9.4) | 0.70 |
| Moderate | | 11 | **-46.6 (-71.3, -2.0)** | 66 | -10.9 (-31.2, 15.4) | 3 | 25.8 (-16.0, 88.4) | 13 | -25.0 (-53.8, 21.6) | 14 | -10.9 (-40.9, 34.3) | 107 | -12.2 (-29.5, 9.4) | 0.20 |
| Severe | | 9 | -12.5 (-55.1, 69.9) | 107 | -13.4 (-28.7, 5.1) | 6 | -35.8 (-75.9, 71.3) | 14 | 29.5 (-20.1, 109.8) | 42 | -2.6 (-32.2, 40.1) | 178 | -8.6 (-21.3, 7.3) | 0.58 |
| p-trend | | | 0.13 | | 0.12 | | 0.44 | | 0.75 | | 0.84 | | 0.63 | |
| None | Model 2[2] | 347 | ref | 494 | ref | 466 | ref | 1037 | ref | 182 | ref | 2526 | ref | |
| Mild | | 19 | -10.0 (-42.9, 41.9) | 147 | 0.0 (-15.7, 18.7) | 30 | 3.4 (-33.8, 61.4) | 33 | -21.6 (-45.8, 13.4) | 41 | -0.6 (-17.5, 19.8) | 270 | -3.0 (-13.1, 8.3) | 0.80 |
| Moderate | | 11 | **-47.4 (-71.5, -2.8)** | 66 | -10.0 (-30.7, 16.7) | 3 | 27.6 (-18.6, 100.1) | 13 | -26.1 (-54.3, 19.5) | 14 | -21.4 (-52.6, 30.4) | 107 | -14.8 (-33.0, 8.3) | 0.20 |
| Severe | | 9 | -10.7 (-54.1, 73.8) | 107 | -15.2 (-30.8, 3.8) | 6 | -27.6 (-72.0, 87.5) | 14 | 23.8 (-24.7, 103.6) | 42 | 0.1 (-22.8, 29.9) | 178 | -7.7 (-20.5, 7.3) | 0.63 |
| p-trend | | | 0.14 | | 0.10 | | 0.76 | | 0.66 | | 0.79 | | 0.05 | |

*(Continued)*

**Table 3.** (Continued)

| | Model | BHS N = 386 n | BHS N = 386 % Difference (95% CI) | BWHS N = 814 n | BWHS N = 814 % Difference (95% CI) | MASALA N = 505 n | MASALA N = 505 % Difference (95% CI) | NHSII N = 1097 n | NHSII N = 1097 % Difference (95% CI) | SHS N = 279 n | SHS N = 279 % Difference (95% CI) | Pooled N = 3081 n | Pooled N = 3081 % Difference (95% CI) | p-het |
|---|---|---|---|---|---|---|---|---|---|---|---|---|---|---|
| None | Model 3[3] | 347 | ref | 494 | ref | 466 | ref | 1037 | ref | 182 | ref | 2526 | ref | |
| Mild | | 19 | -13.7 (-45.2, 35.9) | 147 | 0.3 (-15.6, 19.2) | 30 | 5.8 (-32.7, 66.4) | 33 | -22.6 (-45.8, 10.6) | 41 | -0.6 (-16.0, 17.7) | 270 | -2.0 (-12.2, 8.3) | 0.68 |
| Moderate | | 11 | **-48.9 (-72.4, -5.4)** | 66 | -7.6 (-29.0, 20.4) | 3 | 15.3 (-30.7, 91.9) | 13 | -31.6 (-58.2, 11.9) | 14 | -20.4 (-53.0, 34.7) | 107 | -17.3 (-34.3, 4.1) | 0.26 |
| Severe | | 9 | -4.5 (-51.1, 86.6) | 107 | -15.1 (-30.9, 4.4) | 6 | -25.0 (-72.7, 105.8) | 14 | 23.5 (-27.5, 110.2) | 42 | -2.0 (-23.6, 25.6) | 178 | -8.6 (-21.3, 6.2) | 0.74 |
| p-trend | | | 0.15 | | 0.14 | | 0.81 | | 0.56 | | 0.67 | | 0.12 | |
| **Childhood & Adolescence** | | **n** | **% Difference (95% CI)** | **n** | **% Difference (95% CI)** | **n** | **% Difference (95% CI)** | **n** | **% Difference (95% CI)** | **n** | **% Difference (95% CI)** | **n** | **% Difference (95% CI)** | |
| None | Model 1[1] | 327 | ref | 287 | ref | 427 | ref | 914 | ref | 151 | ref | 2106 | ref | |
| Childhood only | | 20 | 19.6 (-23.8, 87.8) | 207 | -6.8 (-22.1, 11.4) | 39 | -5.0 (-29.2, 27.5) | 123 | 11.4 (-6.2, 32.2) | 31 | -11.5 (-30.3, 12.2) | 420 | 0.0 (-9.5, 10.5) | 0.48 |
| Adolescence only | | 2 | -69.3 (-92.0, 18.1) | 44 | -13.5 (-35.1, 15.2) | 10 | 3.4 (-39.7, 77.2) | 9 | 21.6 (-21.5, 88.3) | 25 | **-27.3 (-42.5, -8.1)** | 90 | -13.1 (-32.3, 10.5) | 0.13 |
| Both | | 37 | -17.4 (-41.4, 16.4) | 276 | -8.9 (-22.5, 7.0) | 29 | -16.1 (-47.8, 34.9) | 51 | -17.4 (-38.7, 11.3) | 72 | 6.6 (-17.4, 37.5) | 465 | -8.6 (-18.1, 2.0) | 0.70 |
| p-trend | | | 0.25 | | 0.25 | | 0.48 | | 0.56 | | 0.93 | | 0.13 | |
| None | Model 2[2] | 327 | | 287 | | 427 | | 914 | | 151 | | 2106 | | |
| Childhood only | | 20 | 17.4 (-25.6, 85.5) | 207 | -7.4 (-22.2, 10.4) | 39 | -4.6 (-27.7, 26.0) | 123 | 13.2 (-4.8, 34.6) | 31 | -15.4 (-37.0, 13.6) | 420 | 0.0 (-9.5, 11.6) | 0.40 |
| Adolescence only | | 2 | -67.9 (-91.7, 23.6) | 44 | -14.3 (-35.8, 14.4) | 10 | 24.3 (-34.0, 134.1) | 9 | 22.6 (-20.5, 89.1) | 25 | **-31.0 (-45.1, -13.4)** | 90 | -13.1 (-34.9, 16.2) | 0.05 |
| Both | | 37 | -17.2 (-41.3, 16.6) | 276 | -9.5 (-23.1, 6.5) | 29 | -8.5 (-41.9, 44.1) | 51 | -17.8 (-38.9, 10.7) | 72 | 5.0 (-17.6, 33.7) | 465 | -8.6 (-18.1, 2.0) | 0.74 |
| p-trend | | | 0.25 | | 0.23 | | 0.81 | | 0.58 | | 0.89 | | 0.14 | |
| None | Model 3[3] | 327 | | 287 | | 427 | | 914 | | 151 | | 2106 | | |
| Childhood only | | 20 | 20.4 (-23.7, 90.2) | 207 | -6.6 (-21.7, 11.3) | 39 | -4.8 (-27.8, 25.7) | 123 | 12.1 (-5.2, 32.6) | 31 | -17.5 (-36.1, 6.6) | 420 | -1.0 (-11.3, 11.6) | 0.29 |
| Adolescence only | | 2 | -71.4 (-92.6, 9.3) | 44 | -11.4 (-33.9, 18.7) | 10 | 20.7 (-35.6, 126.3) | 9 | 11.3 (-31.0, 79.5) | 25 | **-31.9 (-46.2, -13.9)** | 90 | -14.8 (-35.6, 12.7) | 0.09 |
| Both | | 37 | -18.0 (-42.0, 15.7) | 276 | -8.8 (-22.7, 7.6) | 29 | -5.8 (-40.7, 49.6) | 51 | -18.4 (-39.2, 9.5) | 72 | 3.6 (-13.1, 23.5) | 465 | -6.8 (-15.6, 4.1) | 0.60 |
| p-trend | | | 0.23 | | 0.28 | | 0.88 | | 0.48 | | 0.76 | | 0.15 | |

1. Model 1: Adjusted for age.

2. Model 2: Model 1 plus race/ethnicity, gender, mother's educational attainment, father's educational attainment, parental home ownership, loss of parent as a child, childhood financial hardship, childhood public assistance.

3. Model 3: Model 2 plus current BMI, household income, smoking status, physical activity, alternative healthy eating index, depressive symptoms.

**Table 4. Severity of sexual abuse in childhood or adolescence and adult telomere length.**

| | | BHS N = 352 | | | BWHS N = 965 | | | NHSII N = 1097 | | | SHS N = 278 | | | Pooled N = 2678 | | | p-het |
|---|---|---|---|---|---|---|---|---|---|---|---|---|---|---|---|---|---|
| **Childhood (Age <12)** | | n | % Difference (95% CI) | | n | % Difference (95% CI) | | n | % Difference (95% CI) | | n | % Difference (95% CI) | | n | % Difference (95% CI) | | |
| None | Model 1[1] | 342 | Reference | | 626 | Reference | | 842 | Reference | | 242 | Reference | | 2052 | Reference | | |
| Moderate | | 9 | 34.7 (-30.8, 162.2) | | 221 | 9.8 (-5.2, 27.1) | | 128 | 0.3 (-15.0, 18.2) | | 23 | 10.0 (-12.8, 38.8) | | 381 | 7.3 (-3.0, 18.5) | | 0.74 |
| Severe | | 1 | -75.8 (-96.6, 71.6) | | 118 | -11.6 (-28.1, 8.8) | | 127 | **-16.6 (-30.4, -0.1)** | | 13 | -5.5 (-34.6, 36.6) | | 259 | **-13.9 (-23.7, -2.0)** | | 0.56 |
| p-trend | | | | 0.96 | | | 0.61 | | | 0.08 | | | 0.93 | | | 0.15 | |
| None | Model 2[2] | 342 | Reference | | 626 | Reference | | 842 | Reference | | 242 | Reference | | 2052 | Reference | | |
| Moderate | | 9 | 36.3 (-30.2, 166.4) | | 221 | 9.2 (-5.9, 26.7) | | 128 | -0.1 (-15.4, 17.8) | | 23 | -4.0 (-23.1, 19.9) | | 381 | 4.1 (-5.8, 15.0) | | 0.60 |
| Severe | | 1 | -74.8 (-96.4, 76.8) | | 118 | -11.6 (-28.1, 8.7) | | 127 | **-17.4 (-31.1, -0.9)** | | 13 | -21.8 (-45.5, 12.2) | | 259 | **-15.6 (-25.9, -4.9)** | | 0.58 |
| p-trend | | | | 1.00 | | | 0.58 | | | 0.07 | | | 0.19 | | | **0.04** | |
| None | Model 3[3] | 342 | Reference | | 626 | Reference | | 842 | Reference | | 242 | Reference | | 2052 | Reference | | |
| Moderate | | 9 | 40.1 (-28.3, 173.7) | | 221 | 9.3 (-6.1, 27.1) | | 128 | 2.6 (-13.3, 21.4) | | 23 | -6.6 (-24.5, 15.5) | | 381 | 4.1 (-5.8, 15.0) | | 0.49 |
| Severe | | 1 | -74.6 (-96.4, 80.2) | | 118 | -9.7 (-26.4, 10.9) | | 127 | **-17.8 (-31.5, -1.4)** | | 13 | -22.6 (-44.5, 7.9) | | 259 | **-15.6 (-25.9, -4.9)** | | 0.50 |
| p-trend | | | | 0.95 | | | 0.71 | | | 0.07 | | | 0.13 | | | **0.04** | |
| **Adolescence (Age 12–18)** | | n | % Difference (95% CI) | | n | % Difference (95% CI) | | n | % Difference (95% CI) | | n | % Difference (95% CI) | | n | % Difference (95% CI) | | |
| None | Model 1[1] | 341 | Reference | | 633 | Reference | | 855 | Reference | | 241 | Reference | | 2070 | Reference | | |
| Moderate | | 5 | **-68.4 (-86.9, -24.0)** | | 275 | -1.3 (-14.1, 13.4) | | 144 | -2.2 (-17.1, 15.5) | | 31 | 0.3 (-20.7, 26.8) | | 455 | -4.9 (-18.9, 11.6) | | 0.10 |
| Severe | | 0 | | | 57 | -22.0 (-39.6, 0.6) | | 98 | -12.0 (-27.4, 6.6) | | 6 | -16.6 (-60.7, 76.6) | | 161 | **-15.6 (-27.4, -2.0)** | | 0.80 |
| p-trend | | | | | | | 0.17 | | | 0.23 | | | 0.75 | | | 0.18 | |
| None | Model 2[2] | 341 | Reference | | 633 | Reference | | 855 | Reference | | 241 | Reference | | 2070 | Reference | | |
| Moderate | | 5 | **-67.4 (-86.4, -21.7)** | | 275 | -1.8 (-14.9, 13.3) | | 144 | -2.0 (-16.8, 15.5) | | 31 | -12.8 (-32.3, 12.4) | | 455 | -3.9 (-9.5, 3.0) | | 0.10 |
| Severe | | 0 | not estimable | | 57 | -22.1 (-39.8, 0.8) | | 98 | -12.5 (-27.9, 6.0) | | 6 | -30.6 (-63.5, 32.1) | | 161 | **-16.5 (-28.1, -3.0)** | | 0.68 |
| p-trend | | | | | | | 0.15 | | | 0.21 | | | 0.15 | | | 0.08 | |
| None | Model 3[3] | 341 | Reference | | 633 | Reference | | 855 | Reference | | 241 | Reference | | 2070 | Reference | | |
| Moderate | | 5 | **-69.1 (-87.1, -25.7)** | | 275 | -0.4 (-13.8, 15.2) | | 144 | -0.3 (-15.4, 17.6) | | 31 | -14.1 (-35.5, 14.6) | | 455 | -3.9 (-10.4, 4.1) | | 0.06 |
| Severe | | 0 | | | 57 | -19.2 (-37.6, 4.5) | | 98 | -14.1 (-29.6, 4.7) | | 6 | -23.5 (-60.0, 46.4) | | 161 | **-16.5 (-28.1, -3.0)** | | 0.90 |
| p-trend | | | | | | | 0.26 | | | 0.19 | | | 0.23 | | | 0.12 | |
| **Childhood & Adolescence** | | n | % Difference (95% CI) | | n | % Difference (95% CI) | | n | % Difference (95% CI) | | n | % Difference (95% CI) | | n | % Difference (95% CI) | | |
| None | Model 1[1] | 326 | | | 479 | Reference | | 709 | Reference | | 227 | Reference | | 1741 | | | |
| Childhood only | | 7 | 42.2 (-32.6, 200.1) | | 154 | 7.2 (-10.9, 29.0) | | 146 | 0.4 (-16.1, 20.1) | | 14 | -4.4 (-30.0, 30.6) | | 321 | 3.0 (-7.7, 16.2) | | 0.73 |
| Adolescence only | | 3 | **-82.7 (-94.4, -46.4)** | | 147 | -3.2 (-19.5, 16.4) | | 133 | 4.5 (-13.4, 26.1) | | 15 | -16.8 (-42.8, 21.0) | | 298 | -13.1 (-34.3, 15.0) | | 0.02 |
| Both | | 2 | -22.9 (-80.6, 205.9) | | 185 | -3.9 (-18.3, 13.1) | | 109 | **-17.9 (-30.2, -3.4)** | | 22 | 7.8 (-19.2, 43.9) | | 318 | -8.6 (-18.9, 3.0) | | 0.34 |
| p-trend | | | | 0.12 | | | 0.59 | | | 0.17 | | | 0.97 | | | 0.15 | |

*(Continued)*

**Table 4.** (Continued)

| | | BHS N = 352 | | | BWHS N = 965 | | | NHSII N = 1097 | | | SHS N = 278 | | | Pooled N = 2678 | | | p-het |
|---|---|---|---|---|---|---|---|---|---|---|---|---|---|---|---|---|---|
| None | Model 2[2] | 326 | | | 479 | Reference | | 709 | Reference | | 227 | Reference | | 1741 | | | |
| Childhood only | | 7 | 43.9 (-31.8, 203.7) | | 154 | 6.4 (-11.6, 28.0) | | 146 | -0.3 (-16.7, 19.4) | | 14 | -17.5 (-41.1, 15.6) | | 321 | 1.0 (-10.4, 13.9) | | 0.45 |
| Adolescence only | | 3 | **-82.1 (-94.2, -45.0)** | | 147 | -4.1 (-20.6, 15.9) | | 133 | 4.4 (-13.4, 25.8) | | 15 | -26.8 (-52.4, 12.4) | | 298 | -15.6 (-37.5, 12.7) | | 0.01 |
| Both | | 2 | -19.2 (-79.5, 218.4) | | 185 | -4.5 (-19.3, 12.9) | | 109 | **-18.2 (-30.4, -3.8)** | | 22 | -10.6 (-30.1, 14.5) | | 318 | **-11.3 (-20.5, -2.0)** | | 0.62 |
| *p-trend* | | | | 0.14 | | | 0.53 | | | 0.15 | | | 0.12 | | | **0.03** | |
| None | Model 3[3] | 326 | | | 479 | Reference | | 709 | Reference | | 227 | Reference | | 1741 | | | |
| Childhood only | | 7 | 53.0 (-28.1, 225.4) | | 154 | 6.8 (-11.2, 28.6) | | 146 | 2.2 (-14.5, 22.2) | | 14 | -16.8 (-40.2, 15.8) | | 321 | 2.0 (-9.5, 15.0) | | 0.39 |
| Adolescence only | | **3** | **-83.8 (-94.7, -50.0)** | | 147 | -2.1 (-19.1, 18.3) | | 133 | 6.0 (-12.0, 27.6) | | 15 | -22.4 (-49.8, 19.9) | | 298 | -14.8 (-37.5, 16.2) | | 0.01 |
| Both | | 2 | -19.4 (-79.6, 218.4) | | 185 | -2.7 (-17.7, 15.2) | | 109 | **-18.4 (-30.9, -3.6)** | | 22 | -13.2 (-34.5, 14.9) | | 318 | **-11.3 (-20.5, -1.0)** | | 0.53 |
| *p-trend* | | | | 0.12 | | | 0.72 | | | 0.19 | | | 0.15 | | | 0.07 | |

1. Model 1: Adjusted for age.

2. Model 2: Model 1 plus race/ethnicity, gender, mother's educational attainment, father's educational attainment, parental home ownership, loss of parent as a child, childhood financial hardship, childhood public assistance.

3. Model 3: Model 2 plus current BMI, household income, smoking status, physical activity, alternative healthy eating index, depressive symptoms.

abuse in childhood or adolescence though there was variation in prevalence across cohorts. This was higher than national estimates from the 2011–2014 Behavioral Risk Factor Surveillance Survey where 17.9% of respondents reported a history of physical abuse (17.5% of women; 18.4% of men) and 11.6% reported a history of sexual abuse (16.3% of women; 6.7% of men) [55]. We hypothesized that R/S may be a resource for resiliency, but we did not observe evidence that the deleterious impact of sexual abuse was modified by positive or negative religious coping or the extent of religiosity or spirituality in adulthood. Our results demonstrate the long-lasting impact of early-life sexual abuse on health across racial/ethnic groups.

Sexual abuse was more strongly and consistently associated with telomere length than physical abuse. Two recent meta-analyses and one narrative review also concluded that the type of adverse experiences have different impacts on telomere length, though they did not demonstrate differences between sexual and physical abuse [27, 56, 57]. Li et al (2017) found that experiences of childhood separation were associated with shorter telomeres, but physical and sexual abuse and loss of a parent were not [56], while in Ridout et al. (2018), experiences of abuse and neglect were associated with shorter telomeres [27]. Most of the studies included in these reviews did not report estimates for physical and sexual abuse separately. Among those that did, Vincent et al. (2017) found no association between physical or sexual abuse and telomere length [58], while a previous NHSII study found that moderate physical abuse was associated with shorter telomeres, but observed no association with sexual abuse [49]. Our results may reflect severity. Most studies capturing multiple domains of childhood adversity generally assessed the extent of adversity by counting the number of different types of adverse experiences, with each type of adversity measured as present or absent. The previous NHSII study did investigate sexual abuse severity, but our definitions differed, such that some individuals classified as moderate sexual abuse in our study, would have been classified as severe abuse in that publication [49]. Overall, our results suggest that sexual abuse may have long-lasting implications for health.

**Table 5. Joint effects of physical and sexual abuse in childhood or adolescence on adult telomere length.**

| Childhood (Age <12) | | BHS N = 356 | | BWHS N = 787 | | NHSII N = 1097 | | SHS N = 278 | | Pooled N = 2198 | | p-het |
|---|---|---|---|---|---|---|---|---|---|---|---|---|
| | | n | % Difference (95% CI) | n | % Difference (95% CI) | n | % Difference (95% CI) | n | % Difference (95% CI) | n | % Difference (95% CI) | |
| None | Model 1[1] | 295 | Reference | 291 | Reference | 733 | Reference | 167 | Reference | 1486 | Reference | |
| mild physical abuse | | 23 | 13.5 (-25.5, 74.2) | 101 | **-20.1 (-35.4, -1.3)** | 72 | 0.3 (-19.2, 24.3) | 26 | **56.1 (2.8, 137.0)** | 222 | 4.1 (-18.1, 33.6) | 0.04 |
| moderate physical or sexual abuse | | 21 | -12.5 (-43.8, 37.2) | 128 | -5.8 (-23.8, 16.5) | 152 | -6.3 (-20.4, 10.4) | 25 | 6.2 (-13.6, 30.6) | 326 | -3.0 (-13.1, 8.3) | 0.72 |
| severe physical or sexual abuse | | 17 | -14.1 (-47.6, 40.9) | 227 | -3.6 (-18.5, 13.9) | 131 | **-16.4 (-30.0, -0.2)** | 51 | 3.3 (-16.9, 28.4) | 426 | -6.8 (-16.5, 3.0) | 0.47 |
| severe physical and sexual abuse | | 0 | | 40 | -7.3 (-32.6, 27.7) | 9 | 55.1 (-14.1, 180.2) | 9 | -20.7 (-49.6, 24.7) | 58 | -1.0 (-28.1, 36.3) | 0.19 |
| p-trend | | | 0.53 | | 0.76 | | **0.14** | | 0.96 | | 0.25 | |
| None | Model 2[2] | 295 | Reference | 291 | Reference | 733 | Reference | 167 | Reference | 1486 | Reference | |
| mild physical abuse | | 23 | 14.0 (-25.8, 75.1) | 101 | -18.8 (-34.1, 0.0) | 72 | 0.7 (-18.8, 24.9) | 26 | **42.9 (5.2, 94.2)** | 222 | 4.1 (-18.1, 33.6) | 0.02 |
| moderate physical or sexual abuse | | 21 | -11.5 (-43.4, 38.5) | 128 | -7.2 (-25.0, 14.9) | 152 | -6.2 (-20.4, 10.5) | 25 | -8.6 (-27.7, 15.4) | 326 | -3.0 (-13.1, 8.3) | 1.00 |
| severe physical or sexual abuse | | 17 | -12.5 (-46.7, 43.5) | 227 | -3.9 (-18.7, 13.6) | 131 | **-17.3 (-30.7, -1.3)** | 51 | 10.9 (-10.9, 38.0) | 426 | -6.8 (-16.5, 3.0) | 0.23 |
| severe physical and sexual abuse | | 0 | | 40 | -8.4 (-34.4, 28.0) | 9 | 51.6 (-19.0, 183.5) | 9 | **-33.5 (-52.4, -7.0)** | 58 | -1.0 (-28.1, 36.3) | 0.06 |
| p-trend | | | 0.58 | | 0.67 | | **0.12** | | 0.84 | | 0.15 | |
| None | Model 3[3] | 295 | Reference | 291 | Reference | 733 | Reference | 167 | Reference | 1486 | Reference | |
| mild physical abuse | | 23 | 14.1 (-25.8, 75.6) | 101 | -17.9 (-33.4, 1.3) | 72 | 1.5 (-17.8, 25.4) | 26 | 46.0 (11.8, 90.6) | 222 | 7.3 (-16.5, 37.7) | 0.01 |
| moderate physical or sexual abuse | | 21 | -11.7 (-43.8, 38.8) | 128 | -5.5 (-23.6, 17.0) | 152 | -4.1 (-18.5, 12.8) | 25 | -11.1 (-32.1, 16.6) | 326 | -5.8 (-15.6, 6.2) | 0.98 |
| severe physical or sexual abuse | | 17 | -8.8 (-45.2, 51.7) | 227 | -3.4 (-18.5, 14.5) | 131 | **-17.4 (-31.0, -1.1)** | 51 | 6.7 (-8.4, 24.2) | 426 | -4.9 (-15.6, 8.3) | 0.20 |
| severe physical and sexual abuse | | 0 | | 40 | -5.1 (-32.2, 32.8) | 9 | 46.6 (-16.3, 156.7) | 9 | -32.0 (-51.8, -4.0) | 58 | -6.8 (-37.5, 37.7) | 0.05 |
| p-trend | | | 0.67 | | 0.78 | | **0.14** | | 0.65 | | 0.27 | |
| **Adolescence (Age 12–18)** | | n | % Difference (95% CI) | n | % Difference (95% CI) | n | % Difference (95% CI) | n | % Difference (95% CI) | n | % Difference (95% CI) | |
| None | Model 1[1] | 306 | Reference | 424 | Reference | 823 | Reference | 173 | Reference | 1726 | Reference | |
| mild physical abuse | | 17 | -12.3 (-45.7, 41.8) | 84 | -8.0 (-26.1, 14.6) | 19 | -0.8 (-40.1, 64.3) | 32 | 6.8 (-14.8, 33.8) | 152 | -2.0 (-15.6, 12.7) | 0.82 |
| moderate physical or sexual abuse | | 15 | **-54.9 (-73.1, -24.3)** | 163 | -6.2 (-21.3, 11.9) | 147 | -2.0 (-16.8, 15.4) | 29 | 2.7 (-27.4, 45.4) | 354 | -11.3 (-28.8, 9.4) | 0.04 |
| severe physical or sexual abuse | | 9 | -17.0 (-57.0, 60.3) | 94 | -6.5 (-23.5, 14.3) | 104 | -12.8 (-27.6, 5.1) | 40 | -2.4 (-30.4, 37.0) | 247 | -9.5 (-19.7, 3.0) | 0.92 |

*(Continued)*

**Table 5.** (Continued)

| | | BHS N = 356 | | | BWHS N = 787 | | | NHSII N = 1097 | | | SHS N = 278 | | | Pooled N = 2198 | | | p-het |
|---|---|---|---|---|---|---|---|---|---|---|---|---|---|---|---|---|---|---|
| severe physical and sexual abuse | | 0 | | | 22 | -31.2 (-56.0, 7.6) | | 4 | 83.3 (-2.5, 244.5) | | 4 | -1.2 (-63.9, 170.4) | | 30 | 5.1 (-45.7, 105.4) | | 0.05 |
| *p-trend* | | | | **0.02** | | | 0.14 | | | 0.35 | | | 0.97 | | | 0.09 | |
| None | Model 2[2] | 306 | Reference | | 424 | Reference | | 823 | Reference | | 173 | Reference | | 1726 | Reference | | |
| mild physical abuse | | 17 | -10.8 (-44.7, 43.9) | | 84 | -10.0 (-27.5, 11.7) | | 19 | 1.0 (-38.7, 66.7) | | 32 | 3.4 (-14.3, 24.7) | | 152 | -3.0 (17.4, 10.5) | | 0.79 |
| moderate physical or sexual abuse | | 15 | **-54.6 (-72.9, -23.9)** | | 163 | -6.6 (-22.0, 11.8) | | 147 | -1.9 (-16.6, 15.4) | | 29 | -5.9 (-35.5, 37.4) | | 354 | 15.0 (-29.5, 7.3) | | 0.05 |
| severe physical or sexual abuse | | 9 | -14.2 (-55.6, 65.7) | | 94 | -7.4 (-24.8, 14.2) | | 104 | -13.6 (-28.3, 4.3) | | 40 | 1.6 (-21.6, 31.5) | | 247 | -8.6 (-18.9, 3.0) | | 0.80 |
| severe physical and sexual abuse | | 0 | | | 22 | -33.3 (-58.2, 6.4) | | 4 | 76.2 (-12.2, 253.9) | | 4 | -20.2 (-64.4, 79.0) | | 30 | -4.9 (-47.8, 73.3) | | 0.07 |
| *p-trend* | | | | **0.03** | | | 0.12 | | | 0.32 | | | 0.84 | | | 0.06 | |
| None | Model 3[3] | 306 | Reference | | 424 | Reference | | 823 | Reference | | 173 | Reference | | 1726 | Reference | | |
| mild physical abuse | | 17 | -12.3 (-45.6, 41.5) | | 84 | -9.1 (-26.8, 12.9) | | 19 | 0.9 (-36.8, 61.1) | | 32 | 3.8 (-10.7, 20.6) | | 152 | -1.0 (-11.3, 11.6) | | 0.73 |
| moderate physical or sexual abuse | | 15 | **-56.1 (-73.9, -26.2)** | | 163 | -5.4 (-21.1, 13.4) | | 147 | 0.0 (-14.8, 17.5) | | 29 | -7.4 (-35.1, 32.2) | | 354 | -12.2 (-29.5, 9.4) | | 0.04 |
| severe physical or sexual abuse | | 9 | -15.1 (-56.6, 65.9) | | 94 | -6.7 (-24.9, 15.8) | | 104 | -13.8 (-29.1, 4.9) | | 40 | -0.6 (-24.0, 30.0) | | 247 | -8.6 (-19.7, 3.0) | | 0.88 |
| severe physical and sexual abuse | | 0 | | | 22 | -31.8 (-57.6, 9.7) | | 4 | 58.4 (-21.7, 220.5) | | 4 | -11.8 (-55.7, 75.5) | | 30 | -42.3 (-42.3, 55.3) | | 0.15 |
| *p-trend* | | | | **0.02** | | | 0.17 | | | 0.35 | | | 0.77 | | | 0.08 | |

1. Model 1: Adjusted for age.

2. Model 2: Model 1 plus race/ethnicity, gender, mother's educational attainment, father's educational attainment, parental home ownership, loss of parent as a child, childhood financial hardship, childhood public assistance.

3. Model 3: Model 2 plus current BMI, household income, smoking status, physical activity, alternative healthy eating index, depressive symptoms.

With respect to abuse timing, adversity earlier in development has shown greater negative effects on telomere length than those occurring later [27]. However, our sexual abuse finding was of similar magnitude in childhood and adolescence. Previous findings of greater impact of abuse earlier in life may reflect a true impact of abuse experienced in a key developmental period [59, 60], or longer duration of abuse exposure. Abuse duration is not captured well by current measures [57]. We attempted to address this by cross-classifying abuse in childhood and adolescence, which showed, that among individuals that experienced physical or sexual abuse in childhood, around half were also abused in adolescence.

We did not observe evidence of effect modification by the extent of religiosity and spirituality or extent of positive and negative religious coping. This could be due to timing of R/S measurement, choice of R/S measures, and low statistical power. Our assessment of R/S occurred in midlife, and may have been influenced by experiences of abuse [61]. Several resources in childhood and adolescence have been associated with resilience including household stability, school engagement, and caregiver or family support [62–64]. Few resources in adulthood have

**Table 6. Associations of sexual Aabuse in childhood and adolescence and telomere length stratified by religion and spirituality factors.**

| | | Extent of Religion/Spirituality | | | | | Positive Religious Coping | | | | | Negative Religious Coping | | | | |
|---|---|---|---|---|---|---|---|---|---|---|---|---|---|---|---|---|
| | | Not, Slightly or Moderately Religious or Spiritual | | Very Religious or Spiritual | | | At or below median | | Above Median | | | At or below median | | Above Median | | |
| **Childhood (Age <12)** | | n | % Difference (95% CI) | n | % Difference (95% CI) | p | n | % Difference (95% CI) | n | % Difference (95% CI) | p | n | % Difference (95% CI) | n | % Difference (95% CI) | p |
| **BHS** | None | | | | | | 116 | | 210 | | 0.81 | 157 | Reference | 166 | Reference | 0.57 |
| | Moderate | | | | | | 4 | 3.0 (182.9, -62.5) | 5 | 66.5 (301.5, -31.6) | | 8 | 35.0 (163.8, -31.6) | 1 | -36.2 (-91.9, 400) | |
| | Severe | | | | | | 0 | Not estimated | 1 | -81.0 (35.0, -97.3) | | 1 | -82.3 (15.0, -97.3) | 0 | Not estimated | |
| **BWHS** | None | 351 | Reference | 275 | Reference | 0.05 | 354 | Reference | 263 | | 0.23 | 293 | Reference | 324 | Reference | 0.75 |
| | Moderate | 123 | -7.4 (-25.6, 15.2) | 98 | 23.5 (0.8, 51.3) | | 126 | 18.3 (-3.1, 44.4) | 87 | -5.4 (-25.3, 19.7) | | 107 | 13.4 (-10.2, 43.2) | 106 | 6.0 (-13.4, 29.7) | |
| | Severe | 51 | -23.8 (-42.6, 1.2) | 67 | 0.9 (-26.0, 37.6) | | 67 | -5.6 (-28.8, 25.3) | 50 | -21.0 (-41.7, 7.0) | | 52 | -9.8 (-32.7, 20.9) | 65 | -12.9 (-34.8, 16.5) | |
| **NHSII** | None | 476 | | 366 | Reference | 0.58 | 397 | Reference | 375 | | 0.16 | 433 | Reference | 350 | Reference | 0.90 |
| | Moderate | 84 | -6.2 (-29.4, 24.4) | 44 | 3.8 (-15.2, 27.0) | | 66 | -13.7 (-31.3, 8.5) | 52 | 15.0 (-11.5, 49.6) | | 62 | -1.5 (-22.9, 25.8) | 52 | 4.9 (-18.6, 35.1) | |
| | Severe | 80 | -7.1 (-32.8, 28.5) | 47 | **-22.0 (-36.9, -3.7)** | | 68 | **-27.8 (-43.2, -8.2)** | 45 | -4.6 (-31.8, 33.4) | | 65 | **-23.2 (-40.3, -1.3)** | 49 | -20.1 (-40.1, 6.7) | |
| **SHS** | None | 336 | Reference | 147 | Reference | 0.14 | 220 | Reference | 230 | | 0.95 | 267 | Reference | 185 | Reference | 0.31 |
| | Moderate | 38 | 28.0 (-2.5, 68.1) | 16 | -16.5 (-35.9, 8.8) | | 21 | 19.7 (-9.3, 57.9) | 30 | -9.3 (-33.1, 22.9) | | 22 | 0.8 (-26.8, 38.9) | 27 | 7.3 (-25.0, 53.7) | |
| | Severe | 14 | 22.8 (-28.1, 109.9) | 8 | **-43.2 (-60.0, -19.2)** | | 10 | **-43.7 (-61.2, -18.3)** | 11 | -12.3 (-36.3, 20.8) | | 13 | -13.8 (-41.1, 26.1) | 8 | -41.3 (-68.6, 9.8) | |
| **Pooled** | None | 1163 | Reference | 788 | | 0.93 | 1087 | Reference | 1078 | | 0.94 | 1150 | Reference | 1025 | Reference | 0.53 |
| | Moderate | 245 | 3.0 (25.9, -16.5) | 158 | 4.1 (27.1, -15.6) | | 217 | 6.2 (28.4, -11.3) | 174 | 1.0 (17.4, -13.1) | | 199 | 6.2 (23.4, -8.6) | 186 | 5.1 (22.1, -8.6) | |
| | Severe | 145 | -11.3 (11.6, -29.5) | 122 | -22.9 (2.0, -41.7) | | 145 | **-25.9 (-3.0, -43.4)** | 107 | -14.8 (3.0, -28.8) | | 131 | **-18.1 (-1.0, -31.6)** | 122 | **-18.9 (-2.0, -33.6)** | |
| **Adolescence (Age 12–18)** | | n | % Difference (95% CI) | n | % Difference (95% CI) | p | n | % Difference (95% CI) | n | % Difference (95% CI) | p | n | % Difference (95% CI) | n | % Difference (95% CI) | p |
| **BHS** | None | | | | | | 116 | Reference | 209 | Reference | 0.14 | 160 | Reference | 162 | Reference | 0.11 |
| | Moderate | | | | | | 1 | **-90.8 (-36.2, -98.7)** | 4 | -56.8 (15.0, -83.8) | | 2 | -30.2 (161.2, -81.4) | 3 | **-79.4 (-34.3, -93.5)** | |
| | Severe | | | | | | 0 | Not estimated | 0 | Not estimated | | 0 | Not estimated | 0 | Not estimated | |
| **BWHS** | None | 338 | Reference | 295 | Reference | 0.80 | 367 | Reference | 256 | Reference | 0.81 | 295 | Reference | 328 | Reference | 0.54 |
| | Moderate | 157 | -2.2 (-21.0, 21.1) | 118 | 0.1 (-17.5, 21.5) | | 149 | -4.5 (-21.7, 16.4) | 120 | -2.1 (-20.7, 20.9) | | 128 | 8.6 (-11.8, 33.8) | 141 | -10.6 (-27.4, 10.2) | |
| | Severe | 30 | -25.4 (-50.2, 12.0) | 27 | -17.4 (-40.1, 13.9) | | 31 | -19.4 (-41.7, 11.3) | 24 | -28.7 (-54.5, 11.7) | | 29 | -27.3 (-49.2, 4.0) | 26 | -19.6 (-45.1, 17.8) | |
| **NHSII** | None | 483 | Reference | 372 | Reference | **0.01** | 406 | Reference | 379 | Reference | 0.61 | 446 | Reference | 351 | Reference | 0.61 |
| | Moderate | 90 | 11.3 (-12.7, 41.9) | 54 | -12.3 (-29.3, 8.8) | | 70 | -1.4 (-21.9, 24.4) | 60 | -7.7 (-28.2, 18.7) | | 66 | -6.2 (-26.2, 19.2) | 61 | -7.5 (-28.7, 20.0) | |

*(Continued)*

**Table 6.** (Continued)

| | | Extent of Religion/Spirituality | | | | | Positive Religious Coping | | | | | Negative Religious Coping | | | | |
|---|---|---|---|---|---|---|---|---|---|---|---|---|---|---|---|---|
| | | Not, Slightly or Moderately Religious or Spiritual | | Very Religious or Spiritual | | | At or below median | | Above Median | | | At or below median | | Above Median | | |
| | Severe | 67 | 14.3 (-16.2, 55.8) | 31 | **-24.8 (-40.6, -4.8)** | | 55 | -18.2 (-36.9, 6.0) | 33 | 0.1 (-29.2, 41.6) | | 48 | -17.1 (-38.9, 12.5) | 39 | -3.2 (-26.4, 27.4) | |
| SHS | None | 341 | Reference | 147 | Reference | 0.20 | 216 | Reference | 239 | Reference | 0.75 | 267 | Reference | 188 | Reference | 0.90 |
| | Moderate | 39 | 16.8 (-3.4, 41.1) | 19 | -12.5 (-31.0, 10.8) | | 29 | 2.8 (-19.0, 30.6) | 25 | 3.9 (-31.8, 58.3) | | 24 | 5.9 (-18.3, 37.2) | 30 | 20.0 (-12.5, 64.4) | |
| | Severe | 8 | 0.4 (-51.1, 106.4) | 5 | **-42.8 (-65.7, -4.3)** | | 6 | -39.3 (-68.5, 16.9) | 7 | **-27.8 (-53.1, 11.1)** | | 11 | -20.9 (-47.3, 18.9) | 2 | -64.4 (-90.1, 27.8) | |
| Pooled | None | 1162 | Reference | 814 | Reference | 0.12 | 1105 | Reference | 1083 | Reference | 0.91 | 1168 | Reference | 1029 | Reference | 0.98 |
| | Moderate | 286 | 8.3 (23.4, -3.9) | 191 | -7.7 (4.1, -18.1) | | 249 | -3.9 (17.4, -21.3) | 209 | -4.9 (10.5, -18.1) | | 220 | 3.0 (17.4, -10.4) | 235 | -9.5 (-32.3, 20.9) | |
| | Severe | 105 | -3.9 (28.4, -27.4) | 63 | **-25.2 (-10.4, -37.5)** | | 92 | **-20.5 (-3.9, -34.9)** | 64 | -16.5 (5.1, -33.6) | | 88 | **-21.3 (-3.9, -35.6)** | 67 | -13.9 (-34.9, 13.9) | |

Models are adjusted for: Age, race/ethnicity, gender, mother educational attainment, father educational attainment, parental home ownership, loss of parent as a child, childhood financial hardship, childhood public assistance.

p: p-value for interaction between ordinal abuse and R/S indicator variables.

been identified [65]. Our multivariable models demonstrate that adult factors did not meaningfully attenuate associations between sexual abuse, while accounting for childhood socioeconomic status modestly strengthened associations. Lastly, while our total sample size of 3,232 is large in comparison to many studies, we had relatively small numbers of exposed individuals in stratified analyses, limiting our power to detect modest associations.

Our study has several important limitations. DNA extraction methods differed between studies, which can influence RTL measurement [66, 67]. However, extraction methods were the same for all participants within a cohort, and we pooled cohort-specific estimates using meta-analysis, limiting the potential impact of this factor. RTL was assessed using a single measure in mid-life, preventing the estimation of associations between abuse in childhood or adolescence and telomere attrition rate. Additionally, our study evaluated associations with telomere length as a marker of accelerated aging, but other markers including DNA methylation, are important and their examination will enhance our understanding of the biological impact of abuse experienced in early-life [68]. Our group has previously shown that childhood abuse victimization was associated with hypermethlation in *NR3C1* [39]. Ongoing analyses within the Study on Psychosocial Stress, Spirituality, and Health are examining the impact of abuse and other stressful life experiences on DNA methylation within candidate genes in the stress pathway (*HSD11B1*, *HSD11B2*, *NR3C1*, *FKBP5*) and in epigenome-wide association studies. Although we controlled for multiple potential confounders, the possibility of residual confounding remains. Abuse assessment differed between cohorts. We attempted to minimize this heterogeneity through data harmonization. Physical and sexual abuse are sensitive topics and participants' willingness to disclose may differ by how the data are collected (in-person vs. online or paper-based questionnaire), cohort composition (family-based vs. unrelated individuals), and cultural beliefs. However, at least one prospective study has shown that bias is

minimized in recall of traumatic life events [69], and the meta-analysis by Hanssen et al. (2017) did not observe heterogeneity between studies with retrospective recall of abuse vs. objective measures [70]. Yet, despite cohort differences in participant demographic characteristics, exposure assessment and sample processing, we observed consistent associations between severe sexual abuse and shortened telomeres.

In conclusion, our study suggests that sexual abuse in childhood or adolescence is associated with accelerated biological aging as evidenced by decreased telomere length. This association was observed across populations representing multiple racial/ethnic group.

## Author Contributions

**Conceptualization:** Erica T. Warner, Ying Zhang, Alexandra Shields.

**Data curation:** Erica T. Warner, Ying Zhang, Yue Gu, Nicholas D. Spence, Alexandra Shields.

**Formal analysis:** Erica T. Warner, Ying Zhang, Yue Gu, Tâmara P. Taporoski.

**Funding acquisition:** Immaculata DeVivo, Julie R. Palmer, Alka M. Kanaya, Shelley A. Cole, Shelley Tworoger, Alexandra Shields.

**Methodology:** Erica T. Warner, Ying Zhang, Yue Gu, Nicholas D. Spence, Alexandra Shields.

**Project administration:** Erica T. Warner, Ying Zhang, Alexandra Shields.

**Resources:** Alexandre Pereira, Immaculata DeVivo, Julie R. Palmer, Alka M. Kanaya, Shelley A. Cole, Shelley Tworoger, Alexandra Shields.

**Supervision:** Immaculata DeVivo, Julie R. Palmer, Alka M. Kanaya, Shelley A. Cole, Shelley Tworoger, Alexandra Shields.

**Writing – original draft:** Erica T. Warner, Ying Zhang, Yue Gu, Alexandra Shields.

**Writing – review & editing:** Erica T. Warner, Ying Zhang, Yue Gu, Tâmara P. Taporoski, Alexandre Pereira, Immaculata DeVivo, Nicholas D. Spence, Yvette Cozier, Julie R. Palmer, Alka M. Kanaya, Namratha R. Kandula, Shelley A. Cole, Shelley Tworoger, Alexandra Shields.

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
