## [Decision Letter · Decision Letter 0]

22 Jun 2020

PONE-D-20-13222

Physical and sexual abuse in childhood and adolescence and leukocyte telomere length: A pooled analysis of the study on psychosocial stress, spirituality, and health

PLOS ONE

Dear Dr. Warner,

Thank you for submitting your manuscript to PLOS ONE. After careful consideration, we feel that it has merit but does not fully meet PLOS ONE’s publication criteria as it currently stands. Therefore, we invite you to submit a revised version of the manuscript that addresses the points raised during the review process.

We look forward to receiving your revised manuscript.

Kind regards,

Gabriele Saretzki, PhD

Academic Editor

PLOS ONE

Journal Requirements:

2. Please provide additional details regarding participant consent. In the Methods section, please ensure that you have specified (1) whether consent was informed and (2) what type you obtained (for instance, written or verbal). If your study included minors, state whether you obtained consent from parents or guardians. If the need for consent was waived by the ethics committee, please include this information.

Reviewers' comments:

Reviewer's Responses to Questions

**Comments to the Author**

1. Is the manuscript technically sound, and do the data support the conclusions?

Reviewer #1: Partly

2. Has the statistical analysis been performed appropriately and rigorously? 

Reviewer #1: Yes

3. Have the authors made all data underlying the findings in their manuscript fully available?

Reviewer #1: Yes

4. Is the manuscript presented in an intelligible fashion and written in standard English?

Reviewer #1: Yes

5. Review Comments to the Author

Reviewer #1: This is an important study and of much merit.

I do have some concerns about the data analyses that should be addressed.

1. Referencing is not always sufficiently contemporaneous for telomere biology/ageing. Citing reviews from 2007 for this topic is outdated. Please use more recent referencing.

2. Assays of <20% for biomarker CVs are simply not acceptable in the field for data to be considered to be robust. Typical inter and intra assay CVs are below 5% and in my experience typically below 3%. This needs addressed and discussed.

3. Discussion of whether sample collection was equivalent ( e.g. PBLs versus PBMC etc) shoudl be included.

4. Discussion of relative socio-economic position , living density , geo-physical environment and crucially diet and inflammatory statusat blood draw are crucial to providing proper context to assess just how robust data interpretations are.

5. Discussion of epigentic influences shoudl also be included. The authors are encouraged to see Laing et al Eur Child Adolesc Psychiatry. 2019 Apr 9. doi: 10.1007/s00787-019-01329-1.

6. PLOS authors have the option to publish the peer review history of their article (what does this mean?). If published, this will include your full peer review and any attached files.

Reviewer #1: No

---

## [Author Response · Author response to Decision Letter 0]

9 Oct 2020

PONE-D-20-13222: Physical and sexual abuse in childhood and adolescence and leukocyte telomere length: A pooled analysis of the study on psychosocial stress, spirituality, and health

We appreciate the insightful feedback on the manuscript and for the opportunity to revise our manuscript to address it.

Editorial

a. We updated author affiliations and removed key messages to meet format guidelines.

2. Please provide additional details regarding participant consent. In the Methods section, please ensure that you have specified (1) whether consent was informed and (2) what type you obtained (for instance, written or verbal). If your study included minors, state whether you obtained consent from parents or guardians. If the need for consent was waived by the ethics committee, please include this information.

a. On page 4 we now state: “Each cohort study obtained written informed consent from their participants as well as institutional review board approval for cohort maintenance and participation in the SSSH.”

Reviewer 1

1. Referencing is not always sufficiently contemporaneous for telomere biology/ageing. Citing reviews from 2007 for this topic is outdated. Please use more recent referencing.

a. Thank you for your feedback. We previously chose to cite seminal papers that established key concepts related to telomere biology and aging. We have updated the literature review to include more recent references.

2. Assays of <20% for biomarker CVs are simply not acceptable in the field for data to be considered to be robust. Typical inter and intra assay CVs are below 5% and in my experience typically below 3%. This needs addressed and discussed.

a. The CV’s presented in the manuscript we were for the exponentiated T:S ratio. These CV’s tend to be higher than raw CVs because they represent the ratio of a ratio. To address this concern, the laboratory calculated the inter and intra assay CVs and we now present those along with the CV for the exponentiated T:S ratio in the text. The inter and intra assay CV’s are all below 1.0%. The revised text, presented on page 7, is shown below. 

i. The total intra assay coefficients of variation (CV) ranged from 0.27% (NHSII) to 0.3% (MASALA) and the inter assay CVs ranged from 0.33% (MASALA) to 0.77% (SHS). CVs for the exponentiated T:S ratio ranged from 4.7% (SHS) to 10.3% (NHS).

3. Discussion of whether sample collection was equivalent (e.g. PBLs versus Peripheral blood mononuclear cells etc) should be included.

a. We describe the processing and DNA extraction procedures in the Relative Telomere Length section of the methods on page 8 and 9: “Genomic DNA was extracted from peripheral blood leukocytes using the QIAmp (Qiagen) 96-spin blood protocol (BWHS and NHSII), phenol-chloroform standard protocol (SHS and BHS), or sodium dodecylsulfate cell lysis followed by a salt precipitation (MASALA).”

b. According to Tolios et al. (doi.org/10.1371/journal.pone.0143889) DNA extraction methods, but not preanalytic storage conditions and other sample processing methods affect measured telomere length. We acknowledge this in the limitation section on page 13. We state: “DNA extraction methods differed between studies, which can influence RTL measurement. However, extraction methods were the same for all participants within a cohort, and we pooled cohort-specific estimates using meta-analysis, limiting the potential impact of this factor on our results.” 

4. Discussion of relative socio-economic position, living density, geo-physical environment and crucially diet and inflammatory status at blood draw are crucial to providing proper context to assess just how robust data interpretations are.

a. Data on characteristics of the participants’ geo-physical environment or inflammatory status was not available in the Study on Psychosocial Stress, Spirituality, and Health. However, adult characteristics associated with socioeconomic position and inflammation including, body mass index (BMI), household income, smoking status, physical activity, alternative healthy eating index (AHEI), and depressive symptoms are included in Table 1a and 1b, and we adjusted for them in our model 3. From a causal inference perspective, these factors cannot confound the association between abuse in childhood and adolescence and telomere length in adulthood because they occur after the exposure. We position them as potential mediators of the association, which could potentially operate differently across cohorts. However, adjustment for these current characteristics had relatively little impact on the association between abuse in childhood and adolescence and telomere length. Additionally, while there is significant variation between cohorts in these adult characteristics, we observed little evidence of heterogeneity in observed associations across cohorts suggesting that these factors were not effect modifiers.

b. To provide more context for the studies, we moved text that describes the population of each study from supplemental table 1 to the methods section of the manuscript on pages 5-7. We have added text to the discussion on page 22 discuss the different populations data and the impact these differences might be expected to have on our results. We conclude that the consistency of association is evidence of the robustness of our findings. We state: “However, despite cohort differences in participant demographic characteristics, exposure assessment and sample processing, we observed consistent associations between severe sexual abuse and shortened telomeres.”

5. Discussion of epigenetic influences should also be included. The authors are encouraged to see Laing et al Eur Child Adolesc Psychiatry. 2019 Apr 9. doi: 10.1007/s00787-019-01329-1.

a. We have added to the study limitations on p.22 the following text: “Additionally, our study evaluated associations with telomere length as a marker of accelerated aging, but other markers including DNA methylation, are important and their examination will enhance our understanding of the biological impact of abuse experienced in early-life.68 Our group has previously shown that childhood abuse victimization was associated with hypermethlation in NR3C1.39 Ongoing analyses within the Study on Psychosocial Stress, Spirituality, and Health are examining the impact of abuse and other stressful life experiences on DNA methylation within candidate genes in the stress pathway (HSD11B1, HSD11B2, NR3C1, FKBP5) and in epigenome-wide association studies.”

---

## [Editor Report · Decision Letter 1]

14 Oct 2020

Physical and sexual abuse in childhood and adolescence and leukocyte telomere length: A pooled analysis of the study on psychosocial stress, spirituality, and health

PONE-D-20-13222R1

Dear Dr. Warner,

We’re pleased to inform you that your manuscript has been judged scientifically suitable for publication and will be formally accepted for publication once it meets all outstanding technical requirements.

Kind regards,

Gabriele Saretzki, PhD

Academic Editor

PLOS ONE

Additional Editor Comments (optional):

The authors addressed any outstanding issue and clarified Cvs etc for the methodology, included a discussion of limitations etc.
---

## [Editor Report · Acceptance letter]

20 Oct 2020

PONE-D-20-13222R1 

Physical and sexual abuse in childhood and adolescence and leukocyte telomere length: A pooled analysis of the study on psychosocial stress, spirituality, and health 

Dear Dr. Warner:

I'm pleased to inform you that your manuscript has been deemed suitable for publication in PLOS ONE. Congratulations! Your manuscript is now with our production department. 

Kind regards, 

on behalf of

Dr. Gabriele Saretzki 

Academic Editor

PLOS ONE